# Brillouin zone folding driven bound states in the continuum

Wenhao Wang[1,2,3], Yogesh Kumar Srivastava[2,3,4], Thomas CaiWei Tan ◉ [2,3], Zhiming Wang ◉ [1] ✉ & Ranjan Singh ◉ [2,3] ✉

Non-radiative bound states in the continuum (BICs) allow construction of resonant cavities with confined electromagnetic energy and high-quality ($Q$) factors. However, the sharp decay of the $Q$ factor in the momentum space limits their usefulness for device applications. Here we demonstrate an approach to achieve sustainable ultrahigh $Q$ factors by engineering Brillouin zone folding-induced BICs (BZF-BICs). All the guided modes are folded into the light cone through periodic perturbation that leads to the emergence of BZF-BICs possessing ultrahigh $Q$ factors throughout the large, tunable momentum space. Unlike conventional BICs, BZF-BICs show perturbation-dependent dramatic enhancement of the $Q$ factor in the entire momentum space and are robust against structural disorders. Our work provides a unique design path for BZF-BIC-based silicon metasurface cavities with extreme robustness against disorder while sustaining ultrahigh $Q$ factors, offering potential applications in terahertz devices, nonlinear optics, quantum computing, and photonic integrated circuits.

Trapping light in ultra-long radiative lifetime subwavelength structures is critical for applications such as lasers[1,2], optical modulators[3], non-linear optics[4,5], and quantum computing[6,7]. Conventional light-trapping strategies rely on the use of metallic mirrors, total internal reflections (TIR), or photonic bandgaps[8] to prevent outgoing waves. An alternative approach to trap light with infinite lifetimes is using optical bound states in the continuum (BICs), which have been demonstrated in various studies[9–16]. Unlike guided modes (GMs) that lie below the light cone and are forbidden to leak out due to TIR[8], BICs reside inside the radiation continuum yet counter-intuitively do not leak into free space, behaving as embedded dark modes. The non-radiative features of BICs can be attributed to their topological far-field polarization characteristics in the momentum space, where BICs are identified as vortices in the far-field polarization carrying integer topological charges[14,17–19]. The remarkable light-matter interactions and exotic far-field polarization attributes of BICs have been harnessed for sensing and imaging[20,21], lasers[1,22,23], nonlinear enhancement[4,5], and quantum photonics[24].

There are primarily three methods for achieving non-radiative BICs. The first method involves identifying eigenmodes that are forbidden from coupling to the radiation continuum due to symmetry (known as symmetry-protected BICs[10,12,15,16]) or separability (noted as separable BICs[25,26]). The second approach is to adjust the system's parameters to cause destructive interference between multiple leakage channels (noted as tunable BICs or accidental BICs[9,11,13]). Tunable or accidental BICs can be further sub-characterized as Fabry-Pérot BICs, Friedrich–Wintgen BICs, and single-resonance parametric BICs according to different parameter-tuning scenarios. The third method involves using inverse construction methods such as potential engineering, hopping rate engineering, and boundary shape engineering[27]. While most of the research on BICs has focused on symmetry-protected BICs and accidental BICs due to difficulties in experimental realization of separable BICs and inverse construction, Brillouin zone folding (BZF) has recently been used to engineer modes at the edge of the first Brillouin zone (FBZ) into BICs[16,28–33]. By introducing periodic

[1]Institute of Fundamental and Frontier Sciences, University of Electronic Science and Technology of China, Chengdu 610054, China. [2]Division of Physics and Applied Physics, School of Physical and Mathematical Sciences, Nanyang Technological University, Singapore 637371, Singapore. [3]Centre for Disruptive Photonic Technologies, The Photonics Institute, Nanyang Technological University, Singapore 637371, Singapore. [4]Present address: Indian Institute of Technology Hyderabad, Sangareddy, Kandi, Telangana, India. ✉e-mail: zhmwang@uestc.edu.cn; ranjans@ntu.edu.sg

perturbations, GMs located below the light line can be folded into the continuum and potentially serve as BICs.

When a BIC becomes a quasi-BIC in the momentum space, the quality ($Q$) factor decreases quadratically with respect to the distance $k-k_{BIC}$ ($Q \propto 1/(k-k_{BIC})^2$) from the $k_{BIC}$ point, where the BIC emerges with a topological charge ±1. This suggests that high $Q$ resonances only persist in a small region around the BIC in momentum space. In addition, the measured $Q$ factors of quasi-BICs are typically much lower than the theoretical predictions when the system approaches the BIC. This is mainly due to the additional radiation losses induced by the fabrication imperfections or disorder, in addition to intrinsic material loss and finite sample size. To address this issue, a recent proposal is to merge several topological charges in the momentum space to reduce the scattering losses and further improve the $Q$ factor of quasi-BICs[14,19]. However, achieving robust ultrahigh-$Q$ resonances over a large area in wavevector space remains a significant challenge, with potential applications in enhancing nonlinear and quantum effects and scalable lasers over large areas.

Here, we present an approach to achieve disorder-robust and sustainable ultrahigh $Q$ factors throughout the entire momentum space by engineering BZF-induced BICs (BZF-BICs). Specifically, we demonstrate that by utilizing different periodic perturbations, all the five fundamental modes supported by the terahertz photonic crystal (THz-PhC) slabs, located below the light cone at X point, can be folded into Γ point to become BZF-BICs. Unlike conventional BICs that show rapid and perturbation-independent decay in $Q$ factor, BZF-BICs exhibit perturbation-dependent enhancement of ultrahigh $Q$ factor in a large portion of the momentum space, as illustrated in Fig. 1a. Moreover, even when structural disorder is introduced, the $Q$ factor of BZF-BICs remains 10 times higher than that of conventional BICs, indicating robust enhancement. Finally, we fabricate THz-PhC slabs and experimentally demonstrate the controllable evolution features of BZF-BICs' radiation loss in the momentum space.

## Results

A THz-PhC slab (Fig. 1b-i) was designed by patterning a silicon membrane (thickness $t = 200$ μm and relative permittivity $\varepsilon_r = 11.9$) with a rectangular array of circular air holes (periodicity in the $x$ direction $a_1 = 140$ μm and $y$ direction $a_2 = 120$ μm, and radius $r = 40$ μm). The gap between the air holes in the $x$ direction is $L = 60$ μm. Numerical simulations were conducted using the commercial software COMSOL Multiphysics to study the eigenmodes. The transverse electric (TE) band structure along the Γ-X direction is shown in Fig. 1c with hollow blue circles, and the FBZ is presented with a solid black box in the inset of Fig. 1c. The five fundamental bands, labeled TE$_1$ to TE$_5$ by frequency, are below the light line (solid red line) at the X point, and noted as TE$_{m,X}$ ($m = 1$ to 5), indicating that these modes behave as GMs and are localized in the transverse direction across the slab due to TIR. By introducing periodic perturbations, such as changing the distance between every two adjacent air holes by Δ$L$ (noted as "gap perturbation" in Fig. 1b-ii), the periodicity of the PhC in the $x$ direction can be doubled ($a_1 = 280$ μm), allowing access to GMs from free-space excitation and transitioning these non-radiative dark modes into radiative resonances[28,29,34,35]. The selection rules for engineering symmetry-protected BICs by folding high symmetry modes to Γ points in different types of two-dimensional PhC lattices have been reported previously by Overvig et al.[29]. As a result of gap perturbation, several changes occur: i) the FBZ size is reduced by half and is depicted as a dashed black box in the inset of Fig. 1c; ii) the X point of the unperturbed PhC is folded into Γ point, bringing the GMs TE$_{1-5,X}$ into the radiation continuum, which can be seen clearly from the folding of the solid circles on the edges of the plotted bands. The mid-point between Γ and X points of the unperturbed PhC becomes the X point of the gap

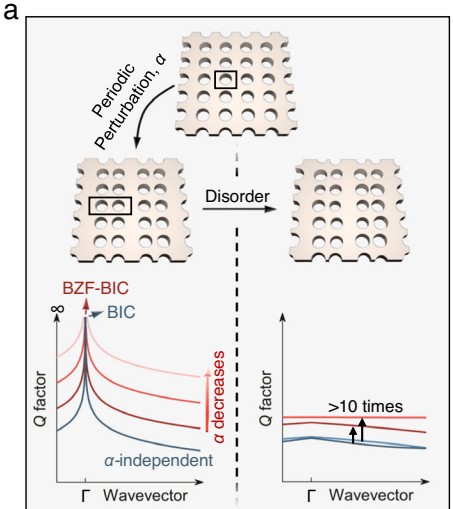

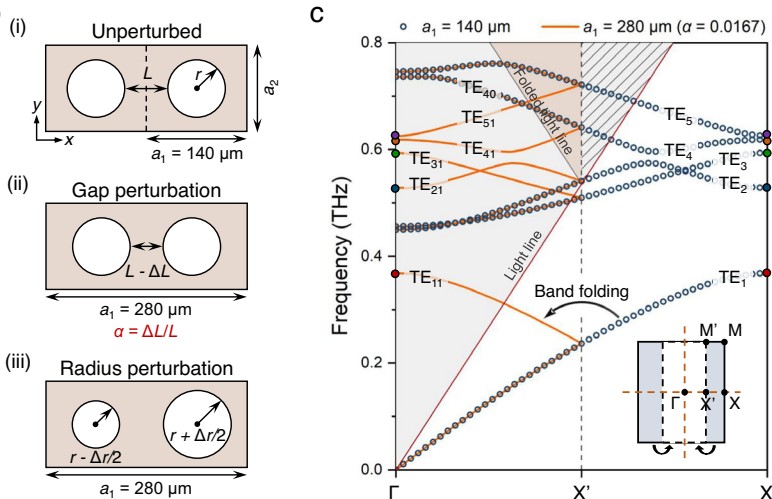

**Fig. 1 | Brillouin zone folding induced bound states in the continuum (BZF-BIC).** **a** Conceptual diagram of BZF-BICs. After introducing periodic perturbation, unit cell size (solid black box) is doubled, and guided modes (GM) are engineered as BZF-BICs. The $Q$ factor of the quasi-BZF-BICs is continuously boosted in the momentum space by decreasing the periodic perturbation. In contrast, the $Q$ factor of the quasi-ordinary BICs does not change. The $Q$ factor enhancement of BZF-BICs compared to ordinary BICs is robust against disorders. **b** Schematic of terahertz photonic crystal (THz-PhC) slabs i) without perturbation, ii) with gap perturbation that the distance between two adjacent air holes in a unit cell is changed by Δ$L$, iii) with radius perturbation that the difference of every two air holes' radius is changed by Δ$r$. Both the gap perturbation and radius perturbation result in a period-doubling in the $x$ direction. **c** Calculated transverse electric (TE) band structures of unperturbed and gap perturbed (perturbation factor $\alpha = 0.0167$, Δ$L = 1$ μm) PhCs,

respectively plotted with hollow blue circles and solid orange lines. The GMs at X point of unperturbed PhC, TE$_{m,X}$ ($m = 1$ to 5), are marked with solid circles of different colors. They are folded into Γ point when gap perturbation is introduced. The bands of gap perturbed PhC are named TE$_{mn}$, where $m$ represents the corresponding band of unperturbed PhC, and $n = 1$ (0) denotes that the bands are obtained with (without) band folding. The solid red line shows the light line of the unperturbed PhC. The shaded gray area without stripe pattern and light-brown area represent the $0^{th}$-order and higher-order diffraction domains of the gap perturbed PhC, respectively. The inset shows that the original first Brillouin zone (FBZ) presented by the solid black box shrinks to its half size after the perturbation is introduced. The bands in the outer half area of unperturbed PhC' FBZ (shaded blue area) are folded into the FBZ of gap perturbed PhC (dashed black box).

perturbed PhC, and to avoid confusion, it's noted as X'. The total supported modes are doubled; iii) the shaded gray area with stripe pattern inside the unperturbed PhC's light cone, where only the zeroth-order diffraction is allowed, is folded into the light-brown area, where higher-order diffractive modes exist. The bands of the gap perturbed PhC with a small perturbation factor $\alpha = 0.0167$ ($\Delta L = 1\,\mu m$), which is defined as $\alpha = \Delta L/L$, are plotted as solid orange lines. They are denoted as $TE_{mn}$, where $m = 1$ to 5 represents the corresponding original band, and $n = 1$ (0) indicates having (no) band folding. Alternatively, the Brillouin zone folding can be achieved by changing the difference in the radius of every two air holes by $\Delta r$ while keeping the gap distance constant ($\Delta L = 0\,\mu m$), which is referred to as "radius perturbation" (Fig. 1b-iii), defined as $\alpha = \Delta r/r$. This perturbation also produces the same band configuration as gap-perturbed PhC.

Next, we study the radiative characteristics of the $TE_{11}$ to $TE_{51}$ folded modes and the $TE_{40}$ unfolded mode. Figure 2a-i illustrates that the $Q$ factor of all the folded modes in the gap perturbed PhC depends on the perturbation factor $\alpha$. Based on the $Q$ factor behavior in the momentum space, the folded modes are classified into two groups: guided resonances (GRs) and BZF-BICs. For the GRs ($TE_{11}$, $TE_{31}$, and $TE_{41}$), their $Q$ factor shows a flat feature in the momentum space and is controlled by the perturbation factor[28,29]

$$Q = Q_0/\alpha^2 \tag{1}$$

where the constant $Q_0$ is determined by the mode, structure design, and material refractive index and remains independent of $\alpha$ for small perturbations. Equation (1) is only applicable when $\alpha$ is smaller than 0.37 for $TE_{11}$ and $TE_{31}$ modes and 0.23 for $TE_{41}$ mode (see Supplementary Section S1). It should be emphasized that although GRs exhibit a similar $Q$ factor evolution pattern to quasi-BICs induced by in-plane inversion ($C_2$) symmetry-breaking at normal incidence[36], their origins differ: GRs emerge from GMs that have no access to radiation channels, whereas quasi-BICs arise from BICs that are surrounded by leakage channels. Furthermore, the perturbations involved are distinct: periodic perturbations are introduced to create GRs, while $C_2$ symmetry-breaking perturbations are necessary to obtain quasi-BICs. The $Q$ factor of BZF-BICs ($TE_{21,\Gamma}$ and $TE_{51,\Gamma}$) approaches infinity at $\Gamma$ point and decreases as one moves away from the BIC in the momentum space

$$Q = Q_0/(\alpha^2 k^2) \tag{2}$$

This suggests that with a small periodic perturbation, the far-field radiation of quasi-BZF-BIC can remain weak even at large $k$ values, indicating an incredibly sustainable ultrahigh $Q$ factor in the momentum space. Further discussion on Eq. (2) is available in Supplementary Section S1. Figure 2a-i shows that at $\alpha = 3.3 \times 10^{-4}$ ($\Delta L = 0.02\,\mu m$), quasi-BZF-BIC $TE_{21,\Delta}$ mode (here $\Delta$ represents the wavevectors between $\Gamma$ and X points) exhibits an ultrahigh $Q$ factor of $8.8 \times 10^9$ for $k_x = 0.1 \times a_1/2\pi$ and $k_y = 0$. This value is six orders of magnitude greater than that of quasi-BIC $TE_{40,\Delta}$ mode, whose $Q$ factor drops to $4.8 \times 10^3$. Even when we consider a perturbation $\alpha = 0.0167$ ($\Delta L = 1\,\mu m$), which is close to the deviations of the fabricated samples (see Supplementary Section S2), the $Q$ factor of quasi-BZF-BIC remains at $3.7 \times 10^6$, which is 770 times higher than that of quasi-BIC.

At the $\Gamma$ point, the unfolded mode $TE_{40}$ has an infinite $Q$ factor which decreases rapidly as it moves away from $\Gamma$ point in the momentum space, following an inverse quadratic relationship with $k$ ($Q \propto 1/k^2$). However, the $Q$ factor of BICs decays independently of $\alpha$ because the mode originates from the unfolded part of the original band, which resides inside the light cone, and band folding has no affect on its radiation into free space. Interestingly, introducing radius perturbation (Fig. 2a-ii) causes the GRs ($TE_{11,\Gamma}$, $TE_{31,\Gamma}$, and

$TE_{41,\Gamma}$) and BZF-BICs ($TE_{21,\Gamma}$ and $TE_{51,\Gamma}$) of the gap perturbed PhC to swap roles with BZF-BICs becoming the GRs of the radius perturbed PhC, but the BIC $TE_{40,\Gamma}$ mode still exhibits perturbation-independent decay in the momentum space. Figure 2b illustrates the $Q$ factor distributions of $TE_{11}$, $TE_{21}$, and $TE_{40}$ modes in the momentum space when a gap perturbation of $\Delta L = 1\,\mu m$ is introduced. The $Q$ factor of GR shows a uniform distribution, while BIC exhibits high $Q$ factor only in a small area around the center of the Brillouin zone. In contrast, BZF-BIC maintains a high $Q$ distribution across a large momentum space.

Figure 2c displays the polarization maps of BZF-BICs and BIC. In the far field of all BICs, vortex centers are evident in the polarization field where the polarization direction cannot be determined. This suggests that BZF-BICs and BICs are decoupled from the radiation continuum at $\Gamma$ point, resulting in a theoretically infinite $Q$ factor. The topological defects in the $TE_{21}$, $TE_{40}$, and $TE_{41}$ modes ($TE_{11}$, $TE_{31}$, and $TE_{51}$ modes) are characterized by an integer topological charge of $q = +1$ ($-1$), which is defined as[17]

$$q = \frac{1}{2\pi} \oint_C d\mathbf{k} \cdot \nabla_\mathbf{k} \phi(\mathbf{k}) \tag{3}$$

where $\phi(\mathbf{k})$ is the angle between the polarization major axis and $x$-axis, and $C$ is a simple closed path in the momentum space that winds around the BIC in the counterclockwise direction. Further information regarding the polarization maps of the six modes studied under gap and radius perturbations can be found in Supplementary Section S3.

Although the $Q$ factor of BICs in infinitely large and perfect PhCs diverges to infinity, it drops to a much lower value in actual samples. Structural disorder is one of the main factors degrading the $Q$ factor. To study the robustness of BZF-BICs, gap perturbed $8 \times 8$ PhC supercells with the disorder are considered. The structure is still assumed to be periodic in the $xy$ plane. The impact of the supercell's size on the $Q$ factor is explored and presented in Supplementary Section S4. Figure 2d shows the consideration of disorder in radius ($dr$) and position ($dx$, $dy$). For each circular air hole, random values of $dr$, $dx$, and $dy$ are chosen from a range of ($-0.8\,\mu m$, $0.8\,\mu m$), ($-0.62\,\mu m$, $0.62\,\mu m$), and ($-0.7\,\mu m$, $0.7\,\mu m$), respectively, which are the average deviations of our fabricated samples from the perfect structure without the disorder (see Supplementary Section S2). The disorder induces coupling of the modes at different $k$ points and affects the coupling of folded modes with leaky channels in the radiation continuum. Specifically, $dr$ and $dx$ contribute to the radius and gap perturbations, respectively. For small $\Delta L$, the $Q$ factor of the folded mode $TE_{21}$ is mainly affected by the radius perturbation induced by $dr$, and it behaves like a GR. Figure 2e shows that when $\Delta L$ is 0.2 and $1\,\mu m$, the $Q$ factor of $TE_{21}$ mode displays a flat distribution in the momentum space, with considerable enhancement compared to the quasi-BIC $TE_{40}$ mode. The dips in $TE_{21}$ mode's $Q$ factor are caused by coupling with other folded modes induced by the periodic boundaries (see Supplementary Section S4). As the supercell size increases, the impact of boundaries becomes negligible, and the simulation model approaches a realistic large-area PhC. For $\Delta L$ of $6\,\mu m$, gap perturbation dominates, and $TE_{21}$ behaves as a quasi-BZF-BIC near $\Gamma$ point. The radiation loss of $TE_{21}$ mode increases at large $k$ values, but its $Q$ factor still displays ~10-fold enhancement over an extensive range of $k$ values compared to the quasi-BIC $TE_{40}$ mode.

To investigate the switching behavior of GRs and BZF-BICs under different perturbations, we analyzed the $C_2$ symmetry of the eigenmodes' field profile. Supplementary Section S5 shows the point group symmetries analysis. In the unperturbed PhC, there are two types of structural high symmetry points based on the structure's periodic symmetry: the center of the air holes and the middle portion of adjacent air holes represented by yellow and green dots in Fig. 3. $TE_{4,\Gamma}$

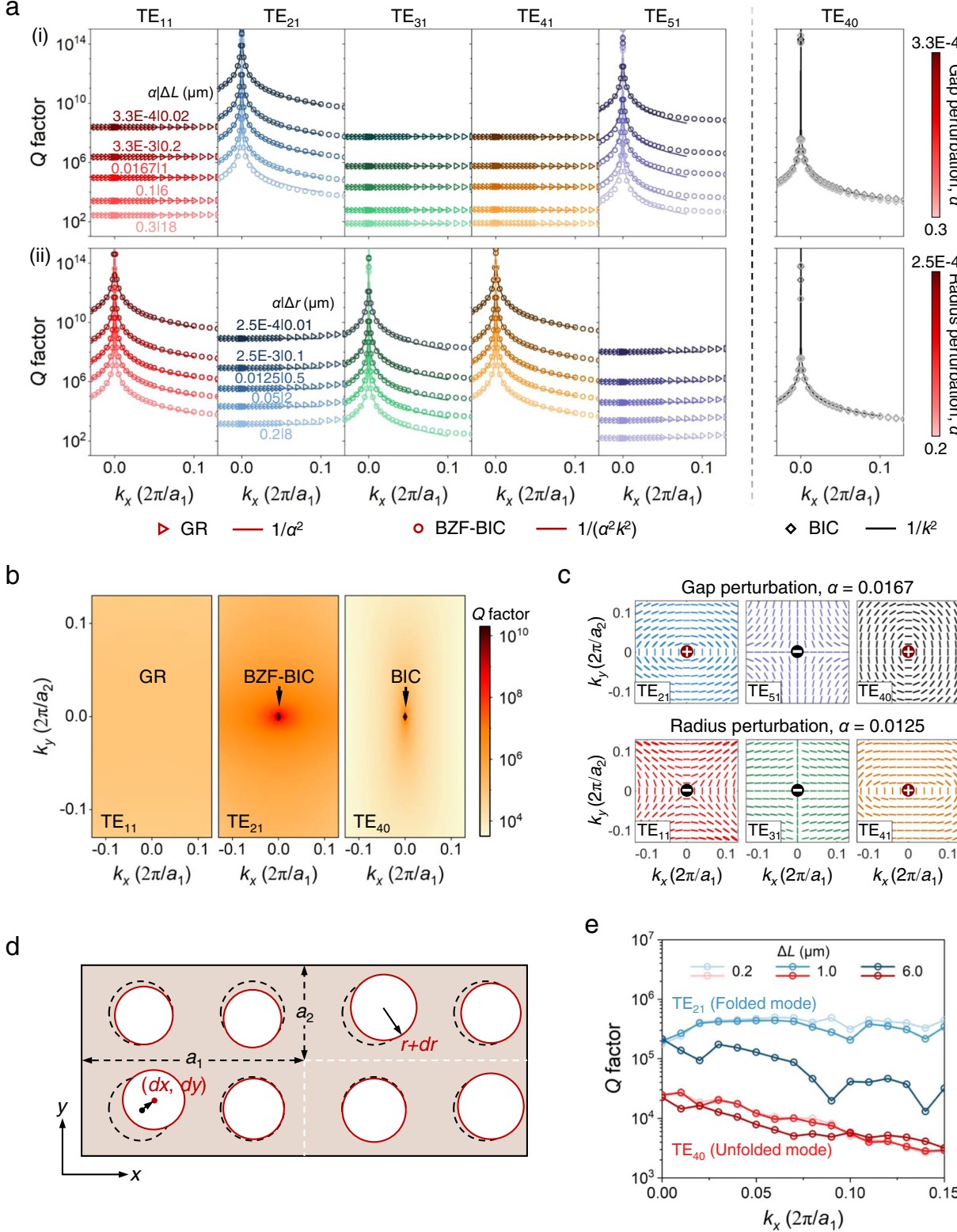

**Fig. 2 | Robust $Q$ factor enhancement and topological properties of BZF-BICs in the momentum space. a** Simulated $Q$ factor of folded (left panel) and un-folded (right panel) bands for different values of i) gap perturbation factor and ii) radius perturbation factor. By introducing different perturbations, folded modes behave as guided resonances (GRs) or BZF-BICs, the $Q$ factor of which evolves following the scaling rule of $Q \propto 1/\alpha^2$ and $Q \propto 1/\alpha^2 k^2$, respectively. The $Q$ factor of BIC (TE$_{40}$) decays quadratically in the momentum space and shows a perturbation-independent feature ($Q \propto 1/k^2$). **b** Simulated $Q$ factor of TE$_{11}$, TE$_{21}$, and TE$_{40}$ modes for a gap perturbed PhC with $\alpha = 0.0167$ ($\Delta L = 1\,\mu m$). Compared to BIC, BZF-BIC shows a sustainable ultrahigh $Q$ factor distribution in a much broader momentum area. **c** Simulated far-field polarization maps of BZF-BICs and BIC for a gap perturbed PhC ($\alpha = 0.0167$, $\Delta L = 1\,\mu m$) and a radius perturbed PhC ($\alpha = 0.0167$, $\Delta r = 0.5\,\mu m$). Each BIC carries a +1 or −1 topological charge at the center of the Brillouin zone. **d** Schematic of a perfect PhC (dashed black circles) and disordered PhC (solid red circles). **e** Simulated $Q$ factors of folded mode TE$_{21}$ and unfolded mode TE$_{40}$ for a gap perturbed PhC with different $\Delta L$ by applying disorder. The simulations were performed in an $8 \times 8$ supercell and were repeated 20 times with randomly generated patterns for averaging. The dips of TE$_{21}$ mode's $Q$ factor are caused by the coupling with other folded modes induced by the periodic boundaries.

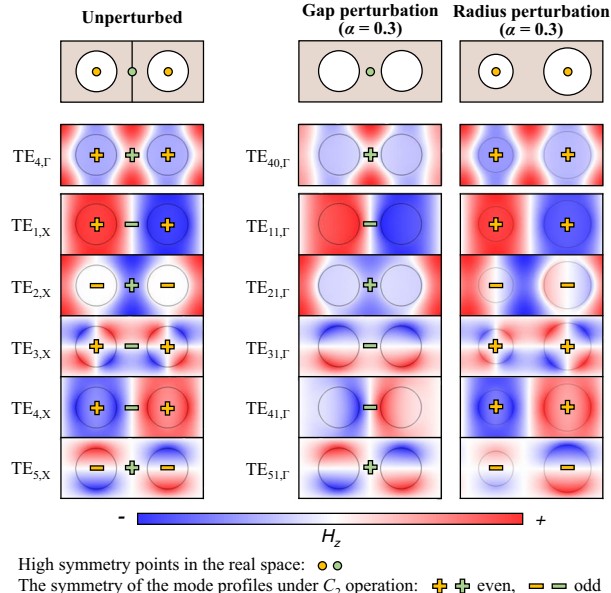

- Unperturbed
- Gap perturbation ($\alpha = 0.3$)
- Radius perturbation ($\alpha = 0.3$)

$H_z$

High symmetry points in the real space: ● ●
The symmetry of the mode profiles under $C_2$ operation: ✚ ✚ even, ▬ ▬ odd

**Fig. 3 | $C_2$ symmetry collapse induced switching of GRs and BZF-BICs.** Calculated magnetic field profiles $H_z$ of eigenmodes at Γ point and X point for unperturbed PhC (left panel), folded and un-folded modes at Γ point for gap perturbed ($\alpha = 0.3$, middle panel) and radius perturbed ($\alpha = 0.3$, right panel) PhCs. Modes at X point for unperturbed PhC show opposite symmetries under $C_2$ operation regarding the high symmetry points at the air holes' center (yellow dots) and at the middle of adjacent air holes (green dots). When a gap (radius) perturbation is introduced, the modes are moved from X point to Γ point, and their $C_2$ symmetry collapses to the one with respect to the middle of adjacent air holes (air holes' center).

**Table 1 | The $C_2$ symmetries and radiating features of eigenmodes in PhCs with and without periodic perturbations**

| Modes | Without perturbation $M^a/C^a$ | Gap perturbation M | Radius perturbation C |
|---|---|---|---|
| TE$_{4,\Gamma}$ / $_{40,\Gamma}$ | +1$^b$/+1(BIC) | +1 (BIC) | +1 (BIC) |
| TE$_{1,X}$ / $_{11,\Gamma}$ | −1$^b$/+1 (GM) | −1 (GR) | +1 (BZF-BIC) |
| TE$_{2,X}$ / $_{21,\Gamma}$ | +1/ −1 (GM) | +1 (BZF-BIC) | −1(GR) |
| TE$_{3,X}$ / $_{31,\Gamma}$ | −1/+1 (GM) | −1(GR) | +1 (BZF-BIC) |
| TE$_{4,X}$ / $_{41,\Gamma}$ | −1/+1 (GM) | −1(GR) | +1 (BZF-BIC) |
| TE$_{5,X}$ / $_{51,\Gamma}$ | +1/−1 (GM) | +1 (BZF-BIC) | −1(GR) |

$^a$M and C denote the high symmetry points of the structure at the middle of adjacent air holes and the center of air holes, respectively.
$^b$+1 and −1 represent even and odd features under the $C_2$ operation.

simulated angle-resolved transmission spectra of THz-PhC ($\Delta L = 30$ μm) along the Γ-X direction under TE and TM polarizations, respectively. Symmetry matching conditions lead to the observation of TE$_{31}$ and TE$_{51}$ modes under the excitation of TM polarized light, where a linear polarized source with one type of symmetry can only excite and couple to the eigenmodes with the same kind of symmetry[37] (see Supplementary Section S6 for more details). The linewidths of GRs TE$_{11}$, TE$_{31}$, and TE$_{41}$ modes remain almost identical as the incident angle increases, indicating that their $Q$ factor shows a flat distribution in the momentum space. In contrast, the linewidths of quasi-BIC TE$_{40,\Delta}$ mode and quasi-BZF-BIC TE$_{21,\Delta}$ and TE$_{51,\Delta}$ modes decrease as the incident angle decreases and ultimately vanish at normal incidence, revealing the evolution from radiating quasi-BIC to non-radiating BIC. The right panel of Fig. 4b, c present the measured transmission spectra obtained through a fiber-coupled photoconductive antenna based terahertz time-domain spectroscopy (THz-TDS) technique (see Methods for more details on the measurement setup), which agrees well with the simulation results.

The experimental results for the THz-PhC samples with different gap perturbations under TE and TM polarizations at normal incidence are presented in Fig. 4d. Due to the $C_2$ symmetry mismatch between the incident wave and eigenmodes, all the BICs are not excited. When the gap perturbation is zero, there is no band folding, and the linewidths of TE$_{11}$, TE$_{31}$, and TE$_{41}$ modes shrink to zero, since they lie below the light line and behave as GMs. After the introduction of gap perturbation, these GMs become GRs, and their coupling with the leakage channels in free space becomes stronger as $\alpha$ increases, which is also observed from the broadening of resonances in the transmission spectra. The $Q$ factor of each GR is extracted by numerically fitting the transmittance spectra, $T(\omega) = |t(\omega)|^2$, to a Fano function $T_F$ on a Lorentzian background $T_d$

$$T(\omega) = T_d + T_F \tag{4}$$

$$T_d = T_0 - I_d \frac{(\gamma_d/2)^2}{(\omega - \omega_d)^2 + (\gamma_d/2)^2} \tag{5}$$

$$T_F = -I_F \frac{(W+q)^2}{(1+q)^2(1+W^2)}, W = \frac{\omega - \omega_F}{\gamma_F/2} \tag{6}$$

where $\omega_F$ ($\omega_d$), $\gamma_F$ ($\gamma_d$), and $I_F$ ($I_d$) are frequency, damping rate, and normalized intensity of GRs (background dipole resonance), $T_0$ is the baseline shift of the whole spectrum, and $q$ is the asymmetry parameter. The $Q$ factor is determined by $Q = \omega_F/2\gamma_F$.

In Fig. 4e, it is shown that as the perturbation factor $\alpha$ approaches 0, the simulated $Q$ factor of GRs diverges to infinity and follows the scaling rule $Q \propto 1/\alpha^2$ (fitted solid lines). Measured $Q$ factors are close to

---

mode shows an even feature under $C_2$ operation for both the yellow and green high symmetry points. However, all the modes at the X point show opposite symmetry for different high symmetry points. For example, TE$_{1,X}$ mode present even and odd features under $C_2$ operation for the yellow and green dots, respectively. When a gap perturbation is introduced, the center of the air holes loses its high symmetry points. The mode symmetry with respect to the middle of adjacent air holes determines the PhC's radiative properties. TE$_{40,\Gamma}$, TE$_{21,\Gamma}$, and TE$_{51,\Gamma}$ are even modes (middle panel of Fig. 3) which have incompatible symmetry with the radiating states whose electric and magnetic vectors are odd under $C_2$ operation and behave as symmetry-protected BICs. TE$_{11,\Gamma}$, TE$_{31,\Gamma}$, and TE$_{41,\Gamma}$ are odd modes acting as radiative GRs. However, when a radius perturbation is introduced (right panel of Fig. 3), TE$_{40,\Gamma}$, TE$_{11,\Gamma}$, TE$_{31,\Gamma}$, and TE$_{41,\Gamma}$ become even modes with respect to the high symmetry points at the air holes' center and become BICs. The relationship between the mode's symmetry and radiative features under different perturbations is summarized in Table 1. The collapse of $C_2$ symmetry induces the switching of GRs and BZF-BICs. When there is no perturbation in the system, the modes located at the X point present both even and odd features under $C_2$ operation with respect to different geometrical high symmetry points. However, after introducing the periodic perturbation, the mode transitions to the Γ point and only shows either even or odd feature under $C_2$ operation with respect to the remaining high symmetry point.

## Experimental demonstration

To demonstrate the exceptional robustness of the ultrahigh $Q$ factor of BZF-BICs in an experimental setting, we fabricate gap perturbed THz-PhCs using photolithography and deep reactive ion etching (DRIE) techniques (see Methods for more details on the fabrication). A SEM image of the sample is presented in Fig. 4a, where $a_1 = 280$ μm, $a_2 = 120$ μm, $r = 40$ μm, $t = 220$ μm, $L = 60$ μm, and $\Delta L = 18$ μm, and the samples are $1 \times 1$ cm² in size. The left panel of Fig. 4b, c show the

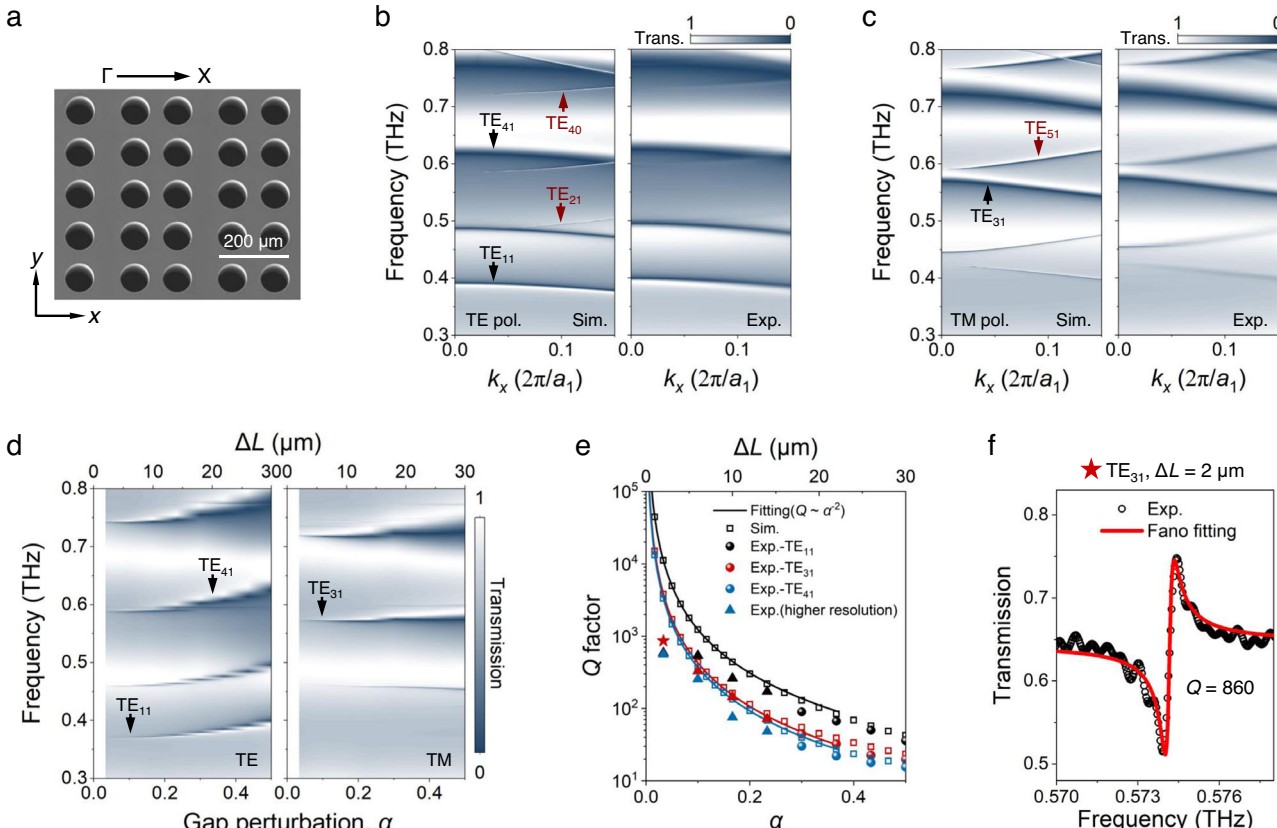

**Fig. 4 | Experimental demonstration of BZF-BICs and GRs. a** Scanning electron microscope (SEM) image of the fabricated gap perturbed ($\Delta L = 18\,\mu m$) THz-PhC sample. Simulated and measured angle-resolved transmission spectra of THz-PhC ($\Delta L = 30\,\mu m$) along the Γ-X direction under (**b**) TE and (**c**) TM polarizations. The linewidths of $TE_{40}$, $TE_{21}$, and $TE_{51}$ modes become narrower as the incident angle decreases, while it's nearly unchanged for GRs $TE_{11}$, $TE_{31}$, and $TE_{41}$ modes.

**d** Measured transmission spectra of THz-PhC samples with different gap perturbations under TE and TM polarizations at normal incidence. **e** Extracted $Q$ factor of GRs $TE_{11}$, $TE_{31}$, and $TE_{41}$ modes from the measured (solid circles) and simulated (hollow squares) transmittance spectra. The simulated $Q$ factor is fitted by $Q \propto 1/\alpha^2$ (solid line). **f** Measured and fitted transmission spectra of $TE_{31}$ mode at normal incidence for $\Delta L = 2\,\mu m$.

the simulated ones at large $\alpha$ values, and higher $Q$ factors of samples with small gap perturbations were obtained using a ZnTe THz-TDS with higher spectral resolution (see Methods for more details on the measurement setup). Transmission spectra obtained by fiber-based, and ZnTe THz-TDSs are similar and shown in Supplementary Section S7. The highest measured $Q$ factor is 860, obtained at $TE_{31}$ mode with $\Delta L = 2\,\mu m$, as depicted in Fig. 4f. Our results are among the highest measured $Q$ factors in terahertz metasurfaces (see Supplementary Section S8). However, there is a discrepancy between measured and simulated $Q$ factors at small perturbations due to three factors: 1) the resolution of the fiber-based and ZnTe THz-TDSs, which is 1.4 GHz and 0.58 GHz, respectively, implies that the maximum measurable $Q$ factor is ~1000 assuming a resonant frequency of 0.6 THz, which is far lower than the simulation results: the simulated $Q$ factor is $1.1 \times 10^4$, $3.8 \times 10^3$, and $3.3 \times 10^3$ for $TE_{11}$, $TE_{31}$, and $TE_{41}$ modes at $\Delta L = 2\,\mu m$, respectively; 2) the diameter of the terahertz beam spot is 8 mm, so the excited mode has a finite lateral size of $S \approx 8$ mm. This finite lateral sized mode consists of a spread of $k$ points with $\delta k_{mode} \approx 2\pi/S \approx 3.5 \times 10^{-2}\,(2\pi/a_1)$; 3) the convergence angle of the incident terahertz beam, which is approximately 6° (see Supplementary Section S9), also leads to a spread of $k$ points, with $\delta k_{source} \approx (2\pi/\lambda)\sin(\theta) \approx 5.2 \times 10^{-2}\,(2\pi/a_1)$. The measured radiative loss is the averaged value within this spread of $k$ points.

To demonstrate the tunable decay characteristics of BZF-BIC radiation loss in the momentum space, we conducted angle-resolved transmission spectra measurements on THz-PhCs with different perturbations. As shown in Fig. 5a, as the value of $\Delta L$ decreases, the

linewidths of all the folded modes become narrower. Specifically, for quasi-BZF-BIC $TE_{51,\Delta}$ mode, its radiation loss relies on both $k$ and the perturbation factor $\alpha$. It shows that by tuning the perturbation, not only an infinite $Q$ factor can be achieved at the Γ point, but also the $Q$ factor of quasi-BZF-BIC can be driven to approach infinity in a large area of momentum space. However, the linewidth of $TE_{40}$ mode in the off-Γ positions remains largely unchanged with the decrease in $\Delta L$, suggesting that the $Q$ factor of conventional BIC displays a perturbation-independent evolution feature in the momentum space (Fig. 5b). In the case of GR $TE_{31}$ mode, its radiation loss only exhibits dependency on $\alpha$. The measured folded modes become weak at small $\alpha$, which is due to the limited scanning length of our fiber-based THz-TDS system. The resonant oscillations beyond 700 ps in the time domain are not captured, and the resonance amplitude becomes very weak. Additionally, the amplitude decreases at oblique incidence due to the degraded collection efficiency of the measurement setup (see Supplementary Section S9), which lowers the measured $Q$ factor at large incident angles.

In conclusion, our findings present a fresh approach to achieve disorder-robust and sustainable ultrahigh $Q$ factor in a significant portion of the momentum space via Brillouin zone folding-bound states in the continuum metasurfaces. Our work establishes the perturbation-dependent evolution and huge enhancement in $Q$ factors of BZF-BICs in the momentum space, which contrasts with the well-established BICs. By introducing different perturbations, we converted all the fundamental guided modes supported in the THz-PhC into BZF-BICs and improved their $Q$ factors significantly. This shows that

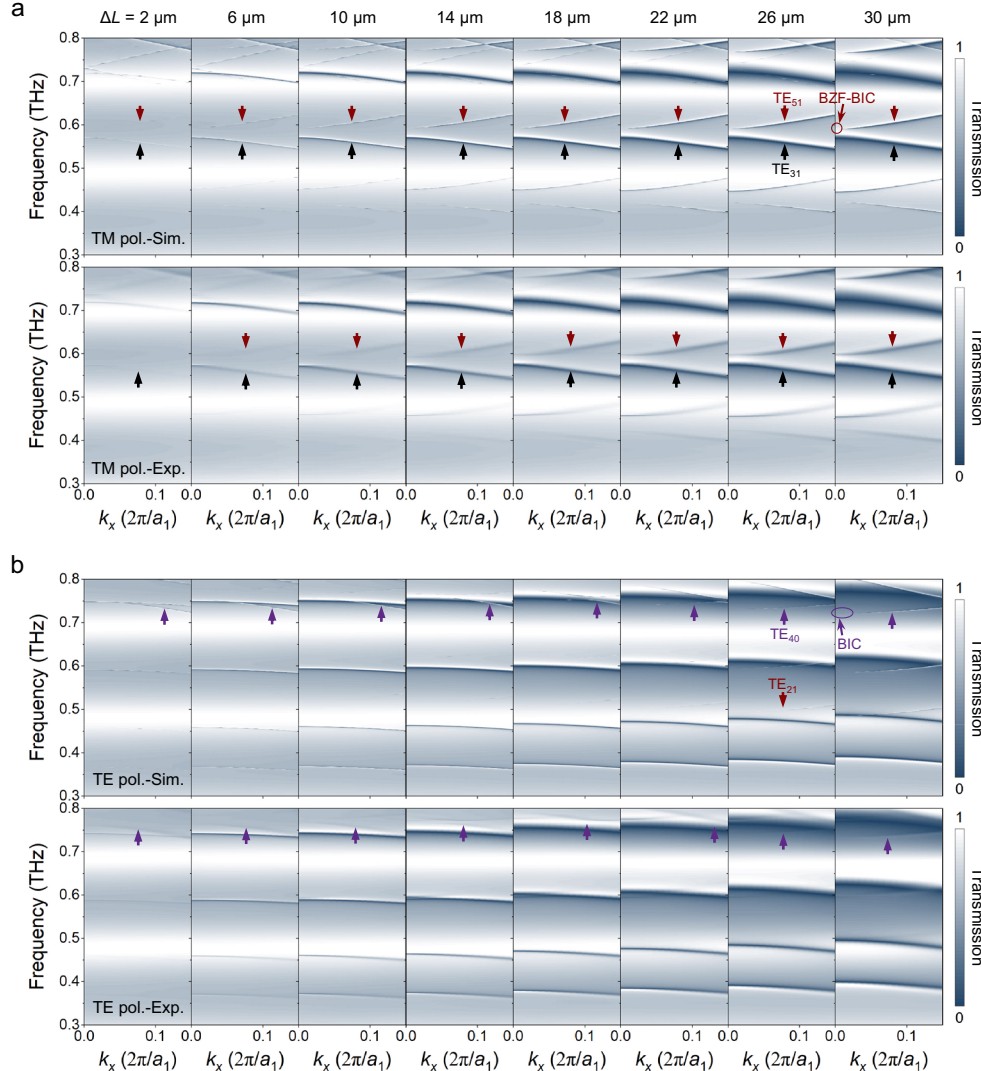

**Fig. 5 | Tunable evolution feature of BZF-BIC's radiation loss in the momentum space.** Simulated (upper panel) and measured (lower panel) angle-resolved transmission spectra of THz-PhCs for $\Delta L = 2$, 6, 10, 14, 18, 22, 26, and 30 μm under (**a**) TM and (**b**) TE polarizations.

Brillouin zone folding provides a universal method for realizing BICs. The enhanced $Q$ factor durability of BZF-BICs over conventional BICs, even under disorder, highlights their potential usefulness in high-$Q$ photonic devices. Our work represents a substantial advancement towards the development of ultra-low threshold, large-area lasers, nonlinear nanophotonic devices, and terahertz cavities that rely on ultrahigh $Q$ factors with exceptional robustness.

## Methods

### Numerical simulation
All simulations were performed using the commercial software COMSOL Multiphysics. The dielectric constant $\varepsilon_r$ was set as 1 and 11.9 for air and Si, respectively. Periodic boundary conditions were applied to the sidewalls of the three-dimensional simulation model. Perfectly matched layers (PML) were added to the top and bottom of the air domain. To study the structure's transmission property, the electromagnetic plane wave was incident from the top boundary. The eigenvalue solver was used to compute the eigenmodes' $Q$ factor and far-field radiation feature.

### Sample fabrication
Samples were fabricated using high-resistivity silicon (>10,000 Ω·cm, 220 μm thick). It was deposited with a 1.5 μm thick $SiO_2$ thin film as an etching protective mask using plasma-enhanced chemical vapor deposition (PECVD). A 1.5 μm layer of AZ5214E photoresist was spin-coated on the $SiO_2$ side of the dioxide-on-silicon (DOS) wafer, which was then patterned by the conventional UV photolithography process. The uncovered area of the $SiO_2$ layer was removed by reactive ion etching (RIE) using mixed gases of $CHF_3$ and $CF_4$. The remaining pattern acted as a protective mask for the subsequent deep reactive ion etching (DRIE, Oxford Estrelas) of the silicon wafer. Each cycle of the Bosch process consisted of two steps: sidewall passivation for 5 s and etching for 15 s. In the deposition step, the $C_4F_8$ gas (85 sccm) was utilized with 600 W ICP power at 35 mTorr pressure. During the etching step, a mixture of $SF_6$ (130 sccm) and $O_2$ (13 sccm) was applied with 600 W ICP power and 30 W bias power at 35 mTorr pressure. The process cycle was repeated until the silicon wafer was etched entirely through.

### THz measurement setup
All the transmission spectra in the main text, except Fig. 4f, were measured using a fiber-based terahertz time-domain spectroscopy. The beam diameter of the terahertz signal passing through the sample was 8 mm. The terahertz transmission signals were scanned for 700 ps, which provides a frequency resolution of 1.4 GHz. Additional zeroes up to 6300 ps were added (zero padding) to the time domain data before

performing the Fourier transform to smoothen the signal spectra through interpolation. We should note that zero padding only increases the number of frequency points with smaller intervals, it does not provide any additional spectral resolution to alter the transmission. Therefore, the actual spectral resolution and the achievable maximum $Q$ factors remain unaltered. Then, the time-domain signals were transformed to frequency domain through the Fourier transform and normalized with the reference air signals to obtain the transmission amplitude. Higher spectral resolution measurements were performed using a ZnTe THz-TDS. A pulsed optical beam was generated from an ultrafast Ti: Sapphire amplifier laser system (800 nm, pulse width 35 fs, and repetition rate 1 kHz) and was used to pump the ZnTe crystal for the generation and detection of terahertz radiation. The scanning time of the terahertz transmission signals is 1734 ps, which provides a frequency resolution of 0.58 GHz. The frequency-domain transmission was then obtained using the same processing method as in the fiber-based THz-TDS.

## Data availability

All the data supporting the findings of this study are openly available in NTU research data repository DR-NTU at https://doi.org/10.21979/N9/XUK1JT. Additional information related to this paper is available from the corresponding author, R.S., upon request.

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

## Acknowledgements

The authors thank P. Agarwal and Y.J. Tan for valuable discussions and suggestions. The authors thank Y.S. Kivshar for fruitful discussions. W.W. and Z.W. acknowledge the support from the National Key Research and Development Program of China (No. 2019YFB2203400) and the "111 Project" (Grant No. B20030). W.W., Y.K.S., T.C.W.T. and R.S. acknowledge funding support from Singapore NRF-CRP23-2019-0005 (TERACOMM). W.W. acknowledges the China Scholarship Council for financial support (202006070143).

## Author contributions

W.W. conceived ideas and initiated the research. W.W. performed simulations and analysis. Y.K.S. fabricated samples. W.W. and T.C.W.T. performed experiment measurements. Z.W. and R.S. co-supervised. W.W. and R.S. wrote the manuscript. R.S. led the overall project.

## Competing interests

The authors declare no competing interests.
