## [Peer Review File · Nature Communications]

Detailed Response to the Referees' Comments

PS: Referees' comments are highlighted in 'red' colour text, authors' responses are shown in 'blue' colour text.

Response to reviewer 1's comments:

Reviewer #1

The authors have nicely addressed all my concerns raised in the last report. I have no further questions.

Response from Authors:

We want to thank the reviewer one last time for their time and constructive feedback.

Response to reviewer #2's comments:

Reviewer #2

The authors have thoroughly addressed my issues and comments, I am happy to recommend it for publication.

Response from Authors:

We want to thank the reviewer one last time for their time and constructive feedback.

Response to reviewer #3's comments:

Reviewer #3:

*The authors did a comprehensive revision and improved the technical quality of the manuscript substantially. However, my previous concerns about novelty of their results were not resolved. In particular, several manuscripts have already explored the physics of high-Q guided and BIC modes originating from the folding of Brillouin zone including Ref [Wang, P., et al Ultra-high-Q resonances in terahertz all-silicon metasurfaces based on bound states in the continuum. *Photonics Research*, 10(12), pp.2743-2750 (2022)] and Ref.[Murai, S., et al. *Engineering Bound States in the Continuum at Telecom Wavelengths with Non-Bravais Lattices. *Laser Photonics Reviews* 16, 2100661 (2022)]. The authors commented that their paper has extra novelty compared to earlier demonstrations, however, all described points are not major enough to fulfil the criteria of novelty of Nature Communications. As I outlined in the first review of this manuscript, I may conclude that the physics and many of essential details of the**

paper are well understood from the previous comprehensive studies, so the manuscript lacks substantial scientific novelty that would make it acceptable for publication in Nature Communications.

Response from Authors:

We thank the reviewer for commenting that we “*did a comprehensive revision and improved the technical quality of the manuscript substantially*”.

The reviewer still questions the scientific novelty of this work. We would like to further clarify this issue in the following.

The main contributions of our work are:

- 1) Our work demonstrated that quasi-BZF-BIC’ Q factor can be considerably enhanced in the momentum space at smaller periodic perturbation.
- 2) Our work demonstrated that the Q factor enhancement of BZF-BIC over ordinary BIC is robust against disorders.
- 3) Our work generalized the idea of using BZF to engineer BIC to all the guided modes in the edge of the first Brillouin zone in photonic crystals with C_{2v} point group symmetry.

Optical BICs have been demonstrated as an alternative scheme to trap light with infinite lifetimes. However, they have been suffering from the dramatic decrease in Q factor when moving away from the BIC point in the momentum space. Our work provides a unique design path for engineering Brillouin zone folding-induced BICs (BZF-BIC), which have extreme robustness against disorder while sustaining ultrahigh Q factors in the momentum space. To the best of our knowledge, none of the novelties of our work has been reported in the existing BZF-BIC related works, including the two references mentioned by the reviewer. Therefore, we believe that our work provides a massive step towards the ultra-low threshold, large-area lasers, and nonlinear nanophotonic devices that require ultrahigh Q factors with extreme robustness.

We hope the above further explanations have resolved the concern of the reviewer.

REVIEWERS' COMMENTS

Reviewer #1 (Remarks to the Author):

The authors have nicely addressed all my concerns raised in the last report. I have no further questions.

Reviewer #2 (Remarks to the Author):

The authors have thoroughly addressed my issues and comments, I am happy to recommend it for publication.

Reviewer #3 (Remarks to the Author):

The authors did a comprehensive revision and improved the technical quality of the manuscript substantially. However, my previous concerns about novelty of their results were not resolved. In particular, several manuscripts have already explored the physics of high-Q guided and BIC modes originating from the folding of Brillouin zone including Ref [Wang, P., et al Ultra-high-Q resonances in terahertz all-silicon metasurfaces based on bound states in the continuum. *Photonics Research*, 10(12), pp.2743-2750 (2022)] and Ref.[Murai, S., et al. Engineering Bound States in the Continuum at Telecom Wavelengths with Non-Bravais Lattices. *Laser Photonics Reviews* 16, 2100661 (2022)]. The authors commented that their paper has extra novelty compared to earlier demonstrations, however, all described points are not major enough to fulfil the criteria of novelty of Nature Communications. As I outlined in the first review of this manuscript, I may conclude that the physics and many of essential details of the paper are well understood from the previous comprehensive studies, so the manuscript lacks substantial scientific novelty that would make it acceptable for publication in Nature Communications.

Detailed Response to the Referees' Comments

PS: Referees' comments are highlighted in 'red' colour text, authors' responses are shown in 'blue' colour text, and the manuscript text are shown in 'black' colour.

Response to reviewer 1's comments:

Reviewer #1

In this work, the authors propose a new type of BIC named dark BIC, which originated from the Brillouin zone folding by periodic perturbation of guided modes. Without perturbation which introduces the band folding, these guided modes are outside the light cone and exhibit an infinite Q factor. When the perturbation is small, the guide modes have a high Q factor. Dark BICs inherit both the high- Q factor from the guided modes and the symmetry-protected BICs, and thus exhibit ultrahigh- Q factors. Compared to the rapid Q -factor-decay feature of isolated BICs (so-called bright BIC) in previous work, the dark BIC presents a perturbation-dependent dramatic enhancement of Q factors even far from the topological center. Besides, the authors provide an experimental demonstration of dark BICs in the sub-THz region. Generally speaking, the manuscript is well-written, and the content is trustworthy. The idea of introducing band-folded guided modes to further increase the Q factor of the dark BICs is quite interesting. However, I cannot recommend the paper in its current form. I will explain my concerns below.

Response from Authors:

We thank the referee for considering our work “well-written”, “trustworthy”, and “quite interesting”. The feedback enabled us to strengthen the manuscript significantly. We will address the referee’s concerns in the following.

Comment #1

Like merging BICs, dark BICs should be robust against perturbations and randomness. Perturbation over α is partly explored, while randomness is not investigated.

Response from Authors:

We thank the reviewer for this critical and insightful comment. Randomness is surely one of the main factors degrading Q factors in real samples. We have performed full-wave electromagnetic simulations to study the robustness of dark BIC, which has been renamed as Brillouin zone folding-induced BIC (BZF-BIC). We have added a new paragraph in the main text to discuss the robustness of BZF-BIC and updated Fig. 2. Additionally, we added two sections to the supplementary information to discuss the disorder in fabricated samples supported with calculated results. Through the simulation investigation, we found that the Q factor of BZF-BIC still shows >10 times enhancement over ordinary BIC even with the onset

of disorder. This demonstrates that the Q factor enhancement of BZF-BIC is robust against disorders. The new text, updated figures, and revised sections in the supplementary are reproduced below for the reviewer's convenience.

Added text in lines 233~257, pages 11~13 of the revised manuscript

Although the Q factor of BICs in infinitely large and perfect PhCs diverges to infinity, it drops to a much lower value in actual samples. Structural disorder is one of the main factors degrading the Q factor. To study the robustness of BZF-BICs, gap perturbed 8×8 PhC supercells with the disorder are considered. The structure is still assumed to be periodic in the xy plane. The effect of the supercell's size on the Q factor is studied and presented in Supplementary Section S4. As shown in Fig. 2d, disorder in radius (dr) and position (dx , dy) has been considered. For each circular air hole, dr , dx , and dy are chosen as random values between $(-0.8 \mu\text{m}, 0.8 \mu\text{m})$, $(-0.62 \mu\text{m}, 0.62 \mu\text{m})$, and $(-0.7 \mu\text{m}, 0.7 \mu\text{m})$, which are the average deviations of our fabricated samples from the perfect structure without the disorder (see Supplementary Section S2). The disorder causes the coupling of the modes at different k points and affect the coupling of the folded modes with the leaky channels in the radiation continuum. Specifically, a disorder in dr and dx contribute to the radius perturbation and gap perturbation, respectively. When ΔL is small, the Q factor of the folded mode TE_{21} is mainly affected by the radius perturbation induced by dr , and TE_{21} mode behaves as GR. As shown in Fig. 2e, when ΔL is 0.2 and 1 μm , the Q factor of TE_{21} shows a flat distribution in the momentum space, exhibiting considerable enhancement than quasi-BIC TE_{40} mode. The dips of the TE_{21} mode's Q factor are caused by the coupling with other folded modes induced by the periodic boundaries (see Supplementary Section S4). The influence of the boundaries will be negligible when the supercell's size is large enough, and the simulation model approaches a realistic large-area PhC. When ΔL is 6 μm , gap perturbation is dominant, and TE_{21} behaves as a quasi-BZF-BIC near Γ point. The radiation loss of TE_{21} mode is increased at large k values. However, the Q factor of TE_{21} mode still shows ~ 10 -fold enhancement than that of quasi-BIC TE_{40} mode over an extensive range of k values.

Fig. 2. Robust Q factor enhancement and topological properties of BZF-BICs in the momentum space. a, Simulated Q factor of folded (left panel) and un-folded (right panel) bands for different values of i) gap perturbation factor and ii) radius perturbation factor. By introducing different perturbations, folded modes behave as guided resonances (GRs) or BZF-BICs, the Q factor of which evolves following the scaling rule of $Q \propto 1/\alpha^2$ and $Q \propto 1/\alpha^2 k^2$, respectively. The Q factor of BIC (TE_{40}) decays quadratically in the momentum space and shows a perturbation-independent feature ($Q \propto 1/k^2$). b, Simulated Q factor of TE_{11} , TE_{21} , and TE_{40} modes for a gap perturbed PhC with $\alpha = 0.0167$ ($\Delta L = 1 \mu m$). Compared to BIC, BZF-BIC shows a sustainable ultrahigh Q distribution in a much broader momentum area. c,

Simulated far-field polarization maps of BZF-BICs and BIC for a gap perturbed PhC ($\alpha = 0.0167$, $\Delta L = 1 \mu\text{m}$) and a radius perturbed PhC ($\alpha = 0.0167$, $\Delta r = 0.5 \mu\text{m}$). Each BIC carries a +1 or -1 topological charge at the center of the Brillouin zone. d, Schematic of a perfect PhC (dashed black circles) and disordered PhC (solid red circles). e, Simulated Q factors of folded mode TE_{21} and unfolded mode TE_{40} for a gap perturbed PhC with different ΔL by applying disorder. The simulations were performed in an 8×8 supercell and were repeated 20 times with randomly generated patterns for averaging. The dips of TE_{21} mode's Q factor are caused by the coupling with other folded modes induced by the periodic boundaries.

Revised Supplementary Section S2: Characterization of disorder in fabricated samples

Samples were fabricated using the conventional UV photolithography process, followed by RIE and DRIE processes (see Methods in the main text). Fig. S2a shows the optical microscopy (OM) image of a gap-perturbed sample ($\Delta L = 18 \mu\text{m}$). The measured diameter is $81.75 \mu\text{m}$, which is $1.75 \mu\text{m}$ larger than the designed value of $80 \mu\text{m}$. We then repeated the measurement over 100 air holes with 3 samples having different ΔL , and summarized the counts of deviations of radius, dr (Fig. S2b). The average dr is estimated to be $0.80 \mu\text{m}$. Similarly, the average deviation of the position of air holes in the x direction dx , and y direction dy are estimated to be $0.62 \mu\text{m}$ and $0.70 \mu\text{m}$, respectively. These values were used in the simulations to study the effect of disorders on eigenmode's Q factor.

Fig. S2. Disorders of fabricated samples. a, Optical microscopy (OM) image of the fabricated gap-perturbed PhC sample ($\Delta L = 18 \mu\text{m}$). Distributions of the deviations of b, air hole's radius dr and locations c, dx and d, dy over 100 air holes with 3 samples having different ΔL .

Supplementary Section S4: Simulation of Q factors for disordered PhC

For actual samples, structural disorders happen arbitrarily, breaking the ideal periodicity of perfect PhC. To study the Q factors of disordered PhC, we assume that the structure is still periodic in an $N \times N$ supercell. The supercell can be regarded as a realistic large-area sample when N is large enough. However, the size of the simulation model increases dramatically with

N , and the simulation time becomes unrealistically long for a large N . To make better trade-off between accuracy and efficiency, we first studied the effect of supercell size on the Q factor. Fig. S4a shows the simulated Q factor of TE_{21} mode using unit cell without disorder and disordered supercells. Gap perturbation $\Delta L = 1 \mu\text{m}$ is applied for all the structures. The Q factor drops as the supercell size N increases, especially in the momentum space near Γ point. The increased radiation loss arises from the enhancement of the coupling between folded mode TE_{21} and leaky channels in the continuum and the coupling between TE_{21} and other folded modes in different k values. It can be noted that the Q factors are similar for disordered 8×8 and 12×12 supercells, suggesting that an 8×8 supercell is enough and reliable to study the Q factors of disordered PhC.

Fig. S4. The effect of supercell size on the Q factors of disordered PhCs. a, Simulated Q factors of TE_{21} mode using unit cell without disorder and using disordered supercells. Gap perturbation $\Delta L = 1 \mu\text{m}$ is applied for all the structures. Simulated eigenmodes supported in b, a unit cell, c, 4×4 , d, 8×8 , and e, 12×12 supercells. Simulated magnetic field profiles H_z of f, $TE_{21,\Gamma}^{1 \times 1}$, g, $TE_{21,\Delta 1}^{4 \times 4}$, h, $TE_{21,\Delta 1}^{8 \times 8}$ and $TE_{21,\Delta 1}^{8 \times 8}$, and i, $TE_{21,\Delta 1}^{12 \times 12}$ and $TE_{21,\Delta 2}^{12 \times 12}$ eigenmodes.

We also notice that there are dips and peaks in the Q factors of disordered PhCs, which changes with the size of the supercell. The fluctuation of the Q factors comes from the coupling with other folded modes induced by the periodic boundaries. As shown in Fig. S4b, TE_{21} band (red dots) is very clean in the band structure of unit cell perfect PhC. The H_z field distribution of BZF-BIC $TE_{21,\Gamma}^{1 \times 1}$ mode shows perfect even symmetry under C_2 operation regarding the middle

of air holes. However, the number of supported eigenmodes increases 15 times in the 4×4 supercell compared to that in the unit cell since the first Brillouin zone (FBZ) is squeezed to its previous $1/16$ and modes previously located outside are folded into the new FBZ. Due to the introduction of disorder, the new folded modes couple with TE_{21} mode and affect its radiation loss (Fig. S4c). The H_z field distribution at the Q factor dip, $TE_{21,\Delta 1}^{4\times 4}$ mode, shows a perturbed feature and presents quasi-even symmetry under C_2 operation regarding the center of supercell (Fig. S4g). As the supercell size increases, the FBZ is further squeezed and modes in a larger portion of the previous momentum space are folded into the new FBZ and coupled with TE_{21} mode.

Comment #2

On line 149, the authors claim that $\Delta L=1\mu\text{m}$ is a large perturbation, which might be true for the sub-THz experiments. However, for more popular optical systems, $\Delta L=1\mu\text{m}$ is only around 0.3% error and far beyond the capabilities of most facilities. This is related to the claims in the abstract, lines 19, 25, the values “nine billion”, “six orders” are vague. The values should be presented more carefully.

Response from Authors:

We thank the reviewer for identifying the potential concerns caused by the way that the results were presented. As shown above, the average deviations of the structure parameters from the ideal case are estimated to be about $0.6\sim 0.8\mu\text{m}$, so $\Delta L=1\mu\text{m}$ is within the fabrication tolerance. However, as the reviewer pointed out, a sample with a perturbation of $\Delta L/P=1\mu\text{m}/280\mu\text{m}=0.36\%$ is challenging to fabricate for the structures working in the optical region. For example, the deviation of the fabricated samples is around 0.9% for photonic crystals with periodicity varying from 530 nm to 580 nm Ref [A1]. We have now modified the text in lines 19~26.

Modified text in lines 19~26, pages 1~2 of the revised manuscript

Here we demonstrate sustainable ultrahigh Q factors in a large tunable portion of the momentum space by engineering Brillouin zone folding-induced BICs (BZF-BICs). All the guided modes are folded into the light cone through periodic perturbation that leads to the emergence of BZF-BICs possessing ultrahigh Q factors throughout the momentum space. Unlike conventional BICs that undergo rapid decay of the Q factor, BZF-BICs show perturbation-dependent dramatic enhancement of the Q factor in the entire momentum space. The enhanced Q factor of BZF-BICs is robust against structural disorders.

Comment #3

For the THz experiments presented in this work, the Q factor is quite small (~ 400) and the capabilities of dark BICs are not fully resolved. To be more specific, the experiments fail to show the advantages of dark BICs (far from the claims in the abstract. At least should be orders higher than other bright BICs as claimed by the authors). Meanwhile, the $1/\alpha^2$ behavior in Fig. 4e is not good (numerical results are perfect).

Response from Authors:

We thank the reviewer for pointing out the concern about the measured Q factor. High Q factor measurement in free space coupled metadevices at terahertz frequencies is difficult and it's mainly limited by the spectral resolution of the terahertz spectroscopy. To improve the measurable maximum Q factor, we used a ZnTe system to do high-resolution measurements. The scanning time of the terahertz transmission signals is extended from 700 to 1734 ps, for which the frequency resolution is enhanced to 0.58 GHz. Thus, the measured maximum Q factor is enhanced from 390 to 860. As shown in Fig. R1, our results are among the highest measured Q factors of the reported free-space coupled THz metasurfaces. Please note that the reported highest Q factor of 1049 in Ref. [A2] is measured using a terahertz frequency-domain spectroscopy system (TeraScan 1550 from Toptica), which provides a frequency resolution of 0.14 GHz.

Fig. R1. High Q factors in THz metasurfaces. a, Q factors of prominent experimental works found in the current literature as compared to our work. They are classified into five categories according to the mechanisms of the resonant modes: GR, quasi-BIC, Fano resonance, electromagnetically induced transparency (EIT), and toroidal resonance. b, Measured transmission spectra of gap-perturbed THz-PhC ($\Delta L = 2 \mu\text{m}$) for TE and TM polarizations. The target GRs are fitted by the Fano formula.

The advantage of BZF-BICs is the enhanced Q factor in the momentum space apart from the Brillouin zone center. From the measured angle-resolved transmission spectra shown below, we can observe that the linewidth of the TE₅₁ mode becomes narrower and finally disappears even at a large k value away from the Brillouin zone center when the gap perturbation approaches zero. However, the linewidth of TE₄₀ mode does not change much and the mode is still observable from the transmission spectra when ΔL decreases to $2 \mu\text{m}$ (Figure R2b), showing a perturbation-independent fast Q -factor-decaying feature. The results suggest that compared to ordinary BIC, the Q factor of BZF-BIC can be enhanced in a large portion of the

momentum space by tuning the periodic perturbation.

Fig. R2. Experimental demonstration of the tunable evolution feature of BZF-BIC's radiation loss in the momentum space. Simulated (upper panel) and measured (lower panel) angle-resolved transmission spectra of THz-PhCs for $\Delta L = 2, 6, 10, 14, 18, 22, 26,$ and $30 \mu\text{m}$ under a, TM and b, TE polarizations.

Although the advantage of BZF-BIC can be qualitatively demonstrated from the transmission spectra, we must admit that we could not experimentally demonstrate it quantitatively, that is to measure several orders higher Q factor of BZF-BIC than that of ordinary BIC. This is mainly due to three reasons: 1) the resolution of our terahertz spectroscopy is 0.58 GHz, which implies that the maximum Q factor we could measure is ~ 1000 if we consider a resonant frequency of 0.6 THz. However, to clearly show the advantage of BZF-BIC, a Q factor of $10^4 \sim 10^5$ needs to be measured; 2) the diameter of the terahertz beam spot is 8 mm, so the excited mode has a finite lateral size of $S \approx 8 \text{ mm}$. This finite-sized mode consist of a spread of k points with δk_{mode}

$\approx 2\pi/S \approx 3.5 \times 10^{-2} (2\pi/a_1)$, where $a_1 = 280 \mu\text{m}$ is the periodicity in the x direction. This could be understood from the fact that the Q factor of quasi-BIC increases with the sample's array size Ref [A3]; 3) the terahertz beam has a convergence angle $\theta \approx 6^\circ$, so the source also has a spread of k points, with $\delta k_{\text{source}} \approx (2\pi/\lambda) \sin(\theta) \approx 5.2 \times 10^{-2} (2\pi/a_1)$. The measured radiative loss will be the averaged value within this spread of k points.

Using the ZnTe system with higher resolution, we re-measured the transmission of gap-perturbed THz-PhC samples under normal incidence. As shown in Fig. R3a and R3b, the measured transmission spectra are very similar using two independent THz spectrometer systems: Fiber coupled photoconductive antenna system and ZnTe non-linear crystal-based system, the spectral resolution of which are 1.4 GHz and 0.58 GHz, respectively. The extracted Q factors are also close for larger ΔL , but show enhancement for the ZnTe system at small ΔL (Fig. R3c). The evolution of GRs' Q factor agrees better with the scaling rule of $Q \propto 1/\alpha^2$. The discrepancy at $\Delta L = 2 \mu\text{m}$ is due to the limitations of our system to measure high Q factors as discussed above and the disorder-induced radiation loss caused by the fabrication imperfection.

Fig. R3. Transmission spectra and Q factors obtained by using two independent terahertz time domain spectrometers, fiber-based and ZnTe based. Measured transmission spectra of THz-PhCs with different gap perturbations under normal incidence by using fiber-based (left panel) and ZnTe (right panel) THz-TDSs for a, TE and b, TM polarizations. c, Simulated and measured Q factors of GRs TE₁₁, TE₃₁, and TE₄₁ modes by using fiber-based and ZnTe THz-TDSs. The simulated Q factor is fitted by $Q \propto 1/\alpha^2$ (solid line).

The discussion above has been added to the manuscript's main text and revised supplementary information and resulted in numerous changes. Fig. R1 and R3 have been added into the supplementary as Fig. S8 and S7, respectively. Fig. R2 is the modified Fig. 5 in the main text.

Comment #4

The critical feature of dark BIC is the slow decay of the Q factor far from the topological center. In this manuscript, the quadratic decay feature $Q=Q_0/(\alpha^2 k^2)$ was provided but lack of proof. I understand it is proven in some other references, but a simple and intuitive demonstration might be better for the readers to follow. Meanwhile, to what extent does the inverse-square law work well? Say, still valid if α approaches 1?

Response from Authors:

We thank the reviewer for bringing up this question. The scaling rule of BZF-BIC's Q factor: $Q \propto 1/\alpha^2 k^2$ could be derived using the combination of coupled mode theory and perturbation theory. To provide an intuitive demonstration, we have added a discussion to Supplementary Section S1, which is reproduced below for the reviewer's convenience.

Revised Supplementary Section S1: Scaling rules of the Q factors of GRs, BICs, and BZF-BICs

Guided resonances (GRs) are formed from guided modes due to periodic perturbation-induced band folding. The Q factors of these leaky modes are related to the perturbation magnitude, α . The relationship between them is shown in Eq. 1 and reproduced here as:

$$Q = Q_0 / \alpha^2, \quad (\text{S1})$$

which can be derived from a combination of perturbation theory and temporal coupled mode theory^[1-2]. As shown in Fig. S1a, the Q factor of GRs TE₁₁, TE₃₁, and TE₄₁ modes at Γ point is well fitted by Eq. S1. From the enlarged Q factor distribution on the right panel of Fig. S1a, we could see that Eq. S1 is valid when α is smaller than 0.37 for TE₁₁ and TE₃₁ modes and 0.23 for TE₄₁ mode. The evolution of symmetry-protected BICs' Q factor in the momentum space follows the rules of^[3-4]

$$Q = Q_0 / k^2, \quad (\text{S2})$$

As shown in Fig. S1b, the Q factor of TE₄₀ mode for a gap-perturbed PhC ($\alpha = 0.0167$, $\Delta L = 1 \mu\text{m}$) is well fitted by Eq. S2 when k is smaller than $0.08 \cdot 2\pi/a_1$.

Since BZF-BIC arises from band folding, the Q factor of quasi-BZF-BIC should also follow the scaling rule of Eq. S1. The left panel of Fig. S1c shows the Q factor of quasi-BZF-BIC TE₂₁ and TE₅₁ modes at $(k_x, k_y) = (0.01 \cdot 2\pi/a_1, 0)$. To exclude the effect of k on the fitting constant Q_0 , we use a modified version of Eq. S1 to fit the Q factor:

$$Q = Q_0 / (0.005^2 \cdot \alpha^2), \quad (\text{S3})$$

As shown in the middle panel of Fig. S1c, the Q factors of TE₂₁ and TE₅₁ modes are fitted well by Eq. S1a when α is smaller than 0.167 for both TE₂₁ and TE₅₁ modes. The fitted Q_0 is 9.02

and 0.185 for TE₂₁ and TE₅₁ modes, respectively. In addition, since BZF-BIC is symmetry-protected BIC, its Q factor should also follow the scaling rule of Eq. S2 in the momentum space. The right panel of Fig. S1c shows the Q factor of quasi-BZF-BIC TE₂₁ and TE₅₁ modes for a gap-perturbed PhC ($\alpha = 0.0167$, $\Delta L = 1 \mu\text{m}$). They are well fitted respectively by

$$Q = 9.02 / (0.0167^2 \cdot k^2), \quad (\text{S4})$$

$$Q = 0.185 / (0.0167^2 \cdot k^2), \quad (\text{S5})$$

when k is smaller than $0.2 \cdot 2\pi/a_1$ for TE₂₁ mode and $0.045 \cdot 2\pi/a_1$ for TE₅₁ mode. From the above mathematical relations, it is easy to conclude that the Q factor BZF-BIC should follow both the rules of Eq. S1 and S2, that is

$$Q = Q_0 / (\alpha^2 k^2). \quad (\text{S6})$$

Fig. S1. Scaling rules of the Q factors of GRs, BICs, and BZF-BICs. a, Simulated Q factors of GRs TE₁₁, TE₃₁, and TE₄₁ modes at Γ point. They are fitted by $Q \propto 1/\alpha^2$ (solid lines). The enlarged Q factor distributions at large α values are shown on the right panel. b, Simulated Q factor of TE₄₀ mode for a gap-perturbed PhC ($\alpha = 0.0167$, $\Delta L = 1 \mu\text{m}$). It is fitted by $Q \propto 1/k^2$. Inset shows the enlarged Q factor distribution at large k_x values. c, Simulated Q factors of quasi-BZF-BIC TE₂₁ and TE₅₁ modes for different α values at $(k_x, k_y) = (0.01 \cdot 2\pi/a_1, 0)$ (left and middle panels) and for different k_x values with a gap-perturbed PhC ($\alpha = 0.0167$, $\Delta L = 1 \mu\text{m}$) (right panel). They are fitted by $Q \propto 1/\alpha^2$ and $Q \propto 1/k^2$, respectively.

From the condition of perturbation theory, we know that the inverse-square law only works

well when the perturbation is small. As discussed above, the $Q \propto 1/\alpha^2$ scaling law for the GRs is valid when α is smaller than ~ 0.3 which changes slightly for different modes. Similarly, the $Q \propto 1/k^2$ scaling law for BICs is valid when k is smaller than $0.08 \cdot 2\pi/a_1$. For the BZF-BICs, the evolution of Q factors follow the $1/\alpha^2 k^2$ scaling law when α is smaller than 0.167 and k is smaller than $0.2 \cdot 2\pi/a_1$ for TE₂₁ mode and $0.045 \cdot 2\pi/a_1$ for TE₅₁ mode. More discussions are added in Supplementary Section S1 reproduced above.

Comment #5

The statement in line 74, “which eventually diverges to infinity in the whole momentum space as the perturbation approaches zero” seems kind of misleading. The eigenstates in the case of zero perturbation are guided modes rather than BICs due to the lack of continuum.

Response from Authors:

We thank the reviewer for pointing out the concerns about the statement. Indeed, when the perturbation decreases to zero, there is no Brillouin zone folding and all the folded modes, including guided resonances and BZF-BICs, transform back to the guided modes and their Q factors become infinite due to total internal reflection. To avoid misleading the readers and to further clarify the difference between ordinary BICs and BZF-BICs, we modified the text in lines 84~88, page 4.

Modified text in lines 84~88, page 4

As distinct from conventional BICs that do not experience band folding caused by a gap or radius perturbations and show rapid Q -factor-decay, BZF-BICs show perturbation-dependent enhancement of ultrahigh Q factor in the large portion of the momentum space, as conceptually illustrated in Fig. 1a.

References

- [A1] Jin, Jicheng, et al. "Topologically enabled ultrahigh-Q guided resonances robust to out-of-plane scattering." *Nature* 574.7779 (2019): 501-504.
- [A2] Wang, Pengfei, et al. "Ultra-high-Q resonances in terahertz all-silicon metasurfaces based on bound states in the continuum." *Photonics Research* 10.12 (2022): 2743-2750.
- [A3] Liu, Zhuojun, et al. "High-Q quasibound states in the continuum for nonlinear metasurfaces." *Physical Review Letters* 123.25 (2019): 253901.

Response to reviewer #2's comments:

Reviewer #2

In this work, the authors experimentally demonstrate BICs and quasi-BICs using a supercell method to increase the robustness of Q -factor with regards to in-plane wavevector ($k_{||}$) compared to using symmetry protected BICs in the first Brillouin zone, these folded modes can achieve improved Q -factors. I think the robustness is neat, but these modes have previously been proposed in theory (doi: 10.1103/PhysRevB.102.035434) and experimentally demonstrated (doi: 10.1038/s41377-022-00905-6, and 10.1515/nanoph-2020-0375). But the characterization of robustness is new and besides the the previously mentioned issues the manuscript is well thought out and organized. Because of this, I think the manuscript requires significant revisions to convey the advancements made more specifically and precisely. Along these lines I have the following comments:

Response from Authors:

We thank the referee for considering our work “neat” and “well thought out and organized”. We also thank the referee’s comments, which enabled us to significantly strengthen the manuscript.

For the advancements of this work, it’s true that the idea of using Brillouin zone folding (BZF) to engineer BICs has been theoretically proposed by Overvig *et al.* Ref[B1], and experimentally demonstrated in Ref[B2~B4] (which has been cited in the main text as Ref[31,32,30]). However, the core feature of our work that reveals the robust ultrahigh Q factors in the momentum space of BZF-induced BIC (BZF-BICs) has not been demonstrated before. In the following, we will justify the novelty of our work in detail from three aspects.

Novelty 1: Quasi-BZF-BIC’ Q factor is considerably enhanced in the momentum space at low periodic perturbation ($Q \propto 1/\alpha^2 k^2$)

For ordinary symmetry-protected BICs, the Q factor of quasi-BICs decreases dramatically when moving away from Γ point in the momentum space (right panel of Fig. R4). However, by decreasing the periodic perturbations, the decay of quasi-BZF-BIC’ Q factor in the momentum space can be slowed down. As shown in the left panel of Fig. R4, when the gap perturbation α decreases, the Q factor of quasi-BZF-BIC in the momentum space is considerably enhanced, following a $1/\alpha^2 k^2$ evolution rule.

Fig. R4. The advancement of BZF-BIC in enhancing Q factor in the momentum space away from Γ point. Simulated Q factor of folded band TE_{21} (left panel) and un-folded band TE_{40} (right panel) for different values of gap perturbation factor α . By decreasing perturbations, BZF-BIC's Q factor evolves following the scaling rule of $Q \propto 1/\alpha^2 k^2$. However, the ordinary BIC's Q factor evolves following the scaling rule of $Q \propto 1/k^2$ and showing gap perturbation independent feature.

Novelty 2: The enhanced Q factor of BZF-BIC is robust against disorder unlike conventional BIC systems

Imperfect fabrication-induced disorder is one of the main factors that degrade BIC's Q factor in actual samples. Thanks to Reviewer #1's Comment 1, we have added the discussions about the effect of disorders on BZF-BIC's and ordinary BIC's Q factors, and explained in detail in response to Comment 1 from Referee #1. Here we provide a summary of the advancement of BZF-BIC on achieving robust high Q factors even in the presence of structural disorders.

In the presence of structural disorder, the Q factors of both ordinary BIC and BZF-BIC degrades. However, the Q factor of BZF-BIC stays more than 10-fold higher than that of ordinary BIC, showing robust enhancement.

Novelty 3: The idea of using BZF to engineer BIC is generalized to all the guided modes at the edge of the first Brillouin zone in photonic crystals with C_{2v} point group symmetry.

Although BZF-BIC has been theoretically and experimentally demonstrated in Ref[B1~B4], and the idea of using different perturbations to transform guided mode (GM) into BZF-BIC or GR has been experimentally demonstrated in Ref[B4], *all the previous works only focus on one or few modes lacking a generalized description for all modes. Here we show that all the five fundamental modes can be engineered into BZF-BICs by introducing gap or radius perturbations.* Through the analysis of the eigenmodes' C_2 symmetries, we revealed that the perturbation-dependent radiation feature of quasi-GM (the GMs that have been folded into the light cone) is related to the C_2 symmetry collapse.

To the best of our knowledge, this is the first time that new features of BZF-BIC have emerged through our work with the generalization of the concept.

To convey the advancements of BZF-BIC more specifically and precisely, we have made numerous changes throughout the manuscript’s abstract, main text, figures, and supplementary information.

In the following, we will address the reviewer’s specific comments point-by-point.

Comment #1

The notation of bright and dark BICs is not consistent with the literature and can create confusion. I think the notation used by Overvig et al. (reference [28] in the manuscript) and table one under the mode notation are more straightforward.

Response from Authors:

We appreciate the reviewer for pointing out the confusion caused by the notation of bright and dark BICs. The notation, $\psi_{L,S}^{m,n}$, used by Overvig *et al.* in Ref[B1] includes detailed information about the modes, including the polarizations of the mode (ψ is TE or TM), the reciprocal lattice point L , the irreducible representation S , the extended zone order m , and the out-of-plane order n . To specify the reciprocal lattice point of the BIC (previously noted as bright BIC) and BZF-BIC (previously noted as dark BIC), all the notation of the modes have been changed to $TE_{mn,L}$, where m is the order of the corresponding original band, $n = 1(0)$ denotes having (no) band folding, and L is the reciprocal lattice point (including Γ , Δ , and X point). To avoid overcomplicating the notation, the extended zone order, the out-of-plane order, and the irreducible representation of the mode are not included. However, the irreducible representation of the modes are studied and presented in **Supplementary Section S5**.

This has resulted in numerous changes throughout the manuscript’s main text and supplementary information. **Supplementary Section S5** is reproduced below for the reviewer’s convenience.

Supplementary Section S5: Point group symmetries analysis

The unperturbed, gap-perturbed, and radius-perturbed PhCs have an in-plane point group symmetry $C_{2v} = \{E, C_2, \sigma_x, \sigma_y\}$, where the symmetry operations are illustrated in Fig. S5. For the unperturbed PhC, there are two types of structural high symmetry points: the center of the air holes and the middle portion of adjacent air holes, respectively represented by the yellow and green dots. However, the radius-perturbed and gap-perturbed PhCs only have the structural high symmetry point in the center of the air holes and the middle portion of adjacent air holes, respectively. For the C_{2v} point groups, the irreducible representations and their characters are listed in Table S1.

Fig. S5. Symmetry operations for the rectangular array of the unperturbed, radius-perturbed, and gap-perturbed PhCs.

Table S1. The character table for the C_{2v} point group

C_{2v}	E	C_2	σ_y	σ_x
A_1	1 ^a	1	1	1
A_2	1	1	-1 ^a	-1
B_1	1	-1	1	-1
B_2	1	-1	-1	1

^aThe number 1 and -1 indicate a symmetric and antisymmetric profile after applying symmetry operations, respectively.

According to the eigenmodes' field profile shown in Fig. 3, the irreducible representation of the modes is identified and summarized in Table S2. For the unperturbed PhCs, $TE_{4,\Gamma}$ mode has both A_1 irreducible representation regarding the center of the air holes and the middle portion of adjacent air holes. However, for all the modes in the X point ($TE_{1-5,X}$), they have different irreducible representations regarding different structural high symmetry points. When gap or radius perturbation is introduced, the folded modes $TE_{(1-5),\Gamma}$ and unfolded mode $TE_{40,\Gamma}$ has the irreducible representation with respect to the remaining structural high symmetry point.

Table S2. The irreducible representations of eigenmodes in PhCs with and without perturbations

Modes	Without perturbation	Gap perturbation	Radius perturbation
	M^a / C^a	M	C
$TE_{4,\Gamma / 40,\Gamma}$	A_1 / A_1	A_1	A_1
$TE_{1,X / 11,\Gamma}$	B_1 / A_1	B_1	A_1
$TE_{2,X / 21,\Gamma}$	A_1 / B_1	A_1	B_1
$TE_{3,X / 31,\Gamma}$	B_2 / A_2	B_2	A_2
$TE_{4,X / 41,\Gamma}$	B_1 / A_1	B_1	A_1
$TE_{5,X / 51,\Gamma}$	A_2 / B_2	A_2	B_2

^aM and C denote the high symmetry points of the structure at the middle of adjacent air holes and the center of air holes, respectively

Comment #2

The process for folding BICs from outside the first Brillouin zone (FBZ) to the FBZ needs to more clearly and explicitly cite previous work done by Overvig et al. (reference [28] in the manuscript) who generated selection rules for all the methods for folding BICs into the FBZ.

Response from Authors:

We appreciate the reviewer pointing out our need for more clarity on explaining the process of folding guided modes (GMs) from outside the FBZ to the FBZ and become BICs. After introducing periodic perturbations, the periodicity of PhC in the x direction doubles. It results in the following consequences: 1) the FBZ shrinks to the half size of the unperturbed PhC's FBZ; 2) the X point of the unperturbed PhCs is folded into the Γ point. It brings the GMs $TE_{1-5,X}$ into the radiation continuum and the number of total supported modes is doubled. To show the folding process more clearly, we added circles in the edges of the plotted bands. 3) the shape of the 0th-order diffraction region of the band diagram changes. To show the folding process more clearly, we have expanded the discussion in lines 112~121, page 6 and reproduced it below.

Modified text in lines 112~121, page 6 of the revised manuscript

As a result, i) the FBZ shrinks to half the size of the unperturbed PhC's FBZ and is plotted with a dashed black box in the inset of Fig. 1c; ii) the X point of the unperturbed PhC is folded into Γ point. It brings the GMs $TE_{1-5,X}$ into the radiation continuum, which can be seen clearly from the folding of the circles on the edges of the plotted bands. The mid-point between Γ and X points of the unperturbed PhC becomes the X point of the gap perturbed PhC, and to avoid confusion, it's noted as X'. The total supported modes are doubled; iii) the shaded gray area with stripe pattern inside the unperturbed PhC's light cone, where only the zeroth-order diffraction is allowed, is folded into the light-brown area, where higher-order diffractive modes exist.

Fig.1c is also modified to show the band folding process more clearly.

Fig. 1. Brillouin zone folding induced bound states in the continuum (BZF-BIC). a, Conceptual diagram of BZF-BICs. After introducing periodic perturbation, unit cell size (solid black box) is doubled, and guided modes (GM) are engineered as BZF-BICs. The Q factor of the quasi-BZF-BICs is continuously boosted in the momentum space by decreasing the periodic perturbation. In contrast, the Q factor of the quasi-ordinary BICs does not change. The Q factor enhancement of BZF-BICs compared to ordinary BICs is robust against disorders. b, Schematic of terahertz photonic crystal (THz-PhC) slabs i) without perturbation, ii) with gap perturbation that the distance between two adjacent air holes in a unit cell is changed by ΔL , iii) with radius perturbation that the difference of every two air holes' radius is changed by Δr . Both the gap perturbation and radius perturbation result in a period-doubling in the x direction. c, Calculated transverse electric (TE) band structures of unperturbed and gap perturbed (perturbation factor $\alpha = 0.0167$, $\Delta L = 1 \mu\text{m}$) PhCs, respectively plotted with hollow blue circles and solid orange lines. The GMs at X point of unperturbed PhC, $\text{TE}_{m,X}$ ($m = 1 \sim 5$), are marked with circles of different colors. They are folded into Γ point when gap perturbation is introduced. The bands of gap perturbed PhC are named TE_{mn} , where m represents the corresponding band of unperturbed PhC, and $n = 1$ (0) denotes that the bands are obtained with (without) band folding. The solid red line shows the light line of the unperturbed PhC. The shaded gray area without stripe pattern and light-brown area represent the 0th-order and higher-order diffraction domains of the gap perturbed PhC, respectively. The inset shows that the original first Brillouin zone (FBZ) presented by the solid black box shrinks to its half size after the perturbation is introduced. The bands in the outer half area of unperturbed PhC' FBZ (shaded blue area) are folded into the FBZ of gap perturbed PhC (dashed black box).

In addition, a discussion about the previous work done by Overvig *et al.* (Ref. [B1]) has been added in lines 107~110, page 5, which can enrich the reader's background of BZF engineering in two-dimensional photonic crystal lattices.

Added text in lines 107~110, page 5 of the revised manuscript

For example, Overvig *et al.* reported the selection rules of engineering symmetry-protected BICs by folding high symmetry modes to Γ points in different types of two-dimensional PhC lattices^[29]

Comment #3

For the THz design these structures are large, and I would have imagined high Q -factors could be achieved, but only a Q -factor of 400 is observed. What is the cause? In the NIR previous work has demonstrated greater than 600 using the BIC Brillouin zone folding technique (doi: 10.1515/nanoph-2020-0375).

Response from Authors:

We thank the reviewer for pointing out the concern about the measured Q factor. This comment is similar to the Comment #3 from Reviewer #1. We have explained in detail in response to the Comment #3 from Reviewer #1. Here we provide a summary of the discussion about the measured Q factors of our work.

- 1) By using a ZnTe THz-TDS with higher resolution, we have improved the measured Q factor from 390 to 860.
- 2) There are mainly three reasons that hinder high Q factor measurement in THz metasurfaces: i) the resolution of terahertz spectroscopy is limited; ii) the excited area of metasurfaces is limited; iii) the excited terahertz beam is not perfectly collimated.
- 3) Although the measured Q factor of our work is less than 1000, which is far lower than the Q factors of NIR BIC metasurfaces (Q factor is $\sim 10^6$ in Ref [B5]). However, our result is among the highest measured Q factors in THz metasurfaces.

References

- [B1] Overvig, Adam C., et al. "Selection rules for quasibound states in the continuum." *Physical Review B* 102.3 (2020): 035434.
- [B2] Malek, Stephanie C., et al. "Multifunctional resonant wavefront-shaping meta-optics based on multilayer and multi-perturbation nonlocal metasurfaces." *Light: Science & Applications* 11.1 (2022): 246.
- [B3] Malek, Stephanie C., et al. "Active nonlocal metasurfaces." *Nanophotonics* 10.1 (2020): 655-665.
- [B4] Murai, Shunsuke, et al. "Engineering Bound States in the Continuum at Telecom Wavelengths with Non-Bravais Lattices." *Laser & Photonics Reviews* 16.11 (2022): 2100661.
- [B5] Chen, Zihao, et al. "Observation of miniaturized bound states in the continuum with ultra-high quality factors." *Science Bulletin* 67.4 (2022): 359-366.

Response to reviewer #3's comments:

Reviewer #3:

In the manuscript, "Dark bound states in the continuum metasurfaces", Wenhao Wang et al. investigate numerically and experimentally resonant optical modes in dielectric metasurfaces composed of equally spaced meta-atoms. The authors study the evolution of resonant properties of modes after band folding associated with dimerizing of meta-atoms and period doubling. They demonstrate that the guided modes (GM) transform into quasi-guided modes which possess a very large radiative Q factor and can become bound states in the continuum (BICs) at the Gamma point depending on the type of perturbation leading to dimerization of the unit cell. The authors investigate numerically how the perturbation type affects the formation of BICs and quasi-GMs, their topological charge and far-field polarization structure. The quasi-GMs with and without BIC features are demonstrated experimentally for the perturbation changing the distance between adjacent meta-atoms. The authors show numerically and experimentally that the Q factor of GMs both with and without BICs at the Gamma point is very robust to the angle of incidence compared to conventional quasi-BICs. The highest measured Q factor of BICs is below 400. The authors associate the predicted GMs with BIC feature as a new type of resonance called "dark BIC".

The idea of engineering of ultrawideband high- Q resonances in dielectric metasurfaces via band folding and dimerizing of meta-atoms is timely and is of high fundamental and practical significance for the field of photonics and optics of subwavelength metastructures. The numerical results of the manuscript are mostly of high technical quality and are supported by solid experimental studies. The paper would make a great candidate for publication in Nature Communications, however, it strongly lacks novelty - an identical study was published earlier this year [see Ref. R1] demonstrating the same effect in dielectric Si metasurfaces in the optical and near-IR range, supported by profound theoretical analysis and measurements. To my regret, all the core ideas of this manuscript, including the physics of "dark BICs", were explored in Ref. [R1], moreover, the Q factor achieved in [R1] is more than 1000 in the near-IR range which exceeds the value achieved in the current manuscript in the THz range. Therefore, I may conclude that now when the physics and many of essential details of the paper are well understood from the previous comprehensive studies, the manuscript lacks substantial scientific novelty that would make it acceptable for publication in Nature Communications. Also, there are some substantial technical flaws in the presented results, that do not affect the main idea but could confuse the readers. Below, I outline a detailed list of comments of criticisms.

Response from Authors:

We thank the referee for commenting that our results have "high technical quality" and are "solid". Since the referee's concern about our work's novelty compared to Ref[R1] is repeated in Comment #1 below, we will address the referee's novelty concern in the response to Comment #1, and other specific comments point-by-point in the following.

Major comments from Reviewer #3.

Comment #1

The core idea of the manuscript is the engineering of high- Q resonances within the whole k -space by band folding of the metasurface resonant modes via dimerization of meta-atoms. A very recent publication [see Ref. R1], which appeared online on 26 Aug 2022 (preprint available on arXiv from 21 Apr 2022), explored the very same idea in an identical structure of Si disks arranged in a square lattice metasurface. The authors of Ref. [R1] did not use the term "dark BIC", but they explored how the band folding effect transforms GM modes into quasi-GMs with a giant Q factor. Moreover, the paper [R1] explores how BICs are formed at the Gamma point for the quasi-GMs achieved through band folding depending on the type of dimerizing perturbation - the authors in [R1] study the same "gap" and "radius" perturbations as in the current manuscript. The reason of why BICs are formed for one quasi-GMs for "gap" perturbation, and for other quasi-GMs for "radius" perturbations are studied in detail in Ref. [R1] with a comprehensive theory behind that. Moreover, Ref. [R1] is an experimental study which confirms all discussed effects in the near-IR and visible range and shows that the Q factor can exceed 1000 at ~ 1240 nm and 300 at ~ 580 nm.

Response from Authors:

This comment from Reviewer #3 questions the novelty/advancement of our work compared to the published Brillouin zone folding induced BIC (BZF-BIC) work Ref [R1]. It is similar to the general remark before Comment #1 from Reviewer #2. We have explained in detail in response to the general remark before Comment #1 from Reviewer #2. Here we provide a summary of the main novelty of our work.

- 1) **Our work demonstrated that quasi-BZF-BIC' Q factor can be considerably enhanced in the momentum space at smaller periodic perturbation.** For the ordinary symmetry-protected BICs, the Q factor decreases dramatically when moving away from Γ point in the momentum space, following $1/k^2$ evolution rule. However, by decreasing the periodic perturbations, the decay of quasi-BZF-BICs' Q factor in the momentum space can be slowed down. The Q factor of quasi-BZF-BICs is considerably enhanced, following a $1/a^2k^2$ evolution rule.
- 2) **Our work demonstrated that the Q factor enhancement of BZF-BIC over ordinary BIC is robust against disorders.** When the structural disorders are considered, the Q factor of quasi-BZF-BIC is still >10 times higher than that of ordinary BIC, showing robust enhancement.
- 3) **Our work generalized the idea of using BZF to engineer BIC to all the guided modes in the edge of the first Brillouin zone in photonic crystals with C_{2v} point group symmetry.** Existing works, including Ref [R1], only focus on one or few modes which loss the generality. Here we show that all the five fundamental modes can be engineered into BZF-BICs by introducing gap or radius perturbations.

We note that Reviewer #3 concerns about the similarity of our work with Ref [R1] about the similar structures (they are not identical since pillar structures are studied in Ref [R1] and membrane design is used in our work) and same perturbations studied. Actually, these structures are very common and the perturbations are conventional, so similar structures and perturbations

have been studied even earlier than Ref [R1], such as Ref [C1, C2]. However, to the best of our knowledge, none of the new features of our work summarized above has been reported in the existing BZF-BIC related works.

Regarding Reviewer #3's concern about the measured Q factor, we have improved our Q factor from 390 to 860 by using a high-resolution ZnTe system. Since our work focuses on THz frequency and Ref [R1] targets on visible and NIR range, it would be fairer to compare them with the highest Q factor reported in their own frequency/wavelength range, respectively. As we have explained in detail in the response to Comment #3 from Referee #1 which also concerns about our measured Q factor, the improved Q factor of 860 in our work is among the highest Q factors in THz metasurfaces. However, for NIR metasurfaces, the Q factor of as high as 1.09×10^6 has been experimentally demonstrated using BIC at ~ 1572 nm in Ref [C3], which is three orders higher than 1000 demonstrated in Ref [R1].

Based on the discussions above, we hope Reviewer #3 can agree on the novelty of our work in demonstrating robust ultrahigh Q factors of BZF-BICs.

Comment #2

The terminology "dark BIC" used by the authors is confusing both physically and mathematically. All modes appearing due to band folding of GMs represent quasi-GMs (or, guided resonances [R2]) due to finite radiation losses. Then, at the Gamma point some of them are BICs with infinite Q factor. Away from the Gamma point such modes represent conventional quasi-BICs. The reason that these quasi-BICs possess a large Q factor in the whole Brillouin zone is due to band folding but not due to the BIC physics - the other GMs without BICs also possess a very high Q factor in the whole k -space. Thus, the observed states are not BICs anywhere except the Gamma point and the core physics behind the high Q factor is not due to BIC nature. Moreover, the BICs are already dark modes, so calling them dark is confusing.

Response from Authors:

We thank the reviewer for pointing out the concerns about the terminology "dark BIC". To avoid misleading readers and causing confusion, we have renamed the "dark BIC" to "Brillouin zone folding induced BIC (BZF-BIC)".

Comment #3

I am very confused by the statement that TM polarized excitation can excite TE modes in metasurface made by the authors and explained in the Supporting Information. Along the high symmetry direction of the k -space, such as Gamma-X for the square lattice the mode polarization in the far-field can be always separated into pure TE and TM. This separation can achieve both for the eigenmodes and the excitation beam. This can be proven if the quasi-guided modes are decomposed into the basis of guided modes of an effective waveguide: the coupling between a TE and TM waveguide modes is proportional to their mode overlap within the unit

cell. It is nonzero only for directions in the k -space that are not aligned with the lattice axes. Therefore, the linearly TE (TM) polarized excitation source cannot excite TM (TE) modes. The explanation made in Supporting Information is very confusing, e.g. it states that TE eigenmodes have a nonzero H_z component of the field. It is true for the near-field, but the excitation properties are related to the far-field and the out-of-plane components of all fields are zero for plane waves. The authors should specify why they observe TE modes for TM excitation in calculations and experiment. Also, it is confusing that TE₃₁ and TE₅₁ modes are not seen for TE excitation - there should be a nonzero signature of them in any case because of nonzero mode overlap.

Response from Authors:

This comment from Reviewer #3 questions that TM polarized excitation can excite TE modes, and the explanation in the supplementary information. It has been experimentally demonstrated and well explained in 2012 in Ref [C4], where Lee *et al.* observed TE-like mode in the E_x (TM) polarized reflectivity spectrum and explained it from the symmetry matching condition. First, the field components of TE and TM polarizations for eigenmode and excitation source are different. As we know, the eigenmodes of every two-dimensional photonic crystal can be classified into two distinct polarizations [Ref C5]: TE modes, in which the electric field is confined to the xy plane (E_x, E_y, H_z), and TM modes, in which the magnetic field is confined to the xy plane (H_x, H_y, E_z). Due to the finite thickness of the photonic crystal, the fields will become mostly TE-like and TM-like when moving away from the mirror plane of the structure in the z direction. However, for the TE polarized excitation, the electric field is perpendicular to the excitation plane. Here the excitation plane is xz plane as illustrated in Fig. S6c, so the electric field is polarized along the y direction: ($H_x, E_y, 0$) for normal incidence and (H_x, E_y, H_z) for oblique incidence. The field components for TM polarized excitation are ($E_x, H_y, 0$) for normal incidence and (E_x, H_y, E_z) for oblique incidence. Second, the excitation of TE eigenmodes with TM polarized source can be understood from the symmetry considerations Ref [C4]: exciting the PhC slab with a source of one type of symmetry results in coupling to the modes of the same type of symmetry only. As shown in Fig. S6d, the H_z of eigenmodes TE_{11,Γ}, TE_{21,Γ}, TE_{41,Γ}, and TE_{40,Γ} all have even symmetry under σ_y operation which changes y to $-y$, while TE_{31,Γ} and TE_{51,Γ} modes have odd symmetry. For TE (H_x) and TM (H_y) polarized sources, the magnetic vector has even and odd symmetries under σ_y operation, respectively. Due to the symmetry matching condition, the TE polarized source excite eigenmodes TE_{11,Γ}, TE_{21,Γ}, TE_{41,Γ}, and TE_{40,Γ}, and TM polarized source excite eigenmodes TE_{31,Γ} and TE_{51,Γ}.

Some of the reviewer's statements are not consistent, and are difficult to understand. For example, the reviewer states that "*there should be a nonzero signature of them (TE_{31,Γ} and TE_{51,Γ} modes) in any case because of nonzero mode overlap*". Since mode overlap is only about near field, from the above statement we can conclude that the reviewer think that the mode excitation and observation is related to the near field mode overlap. However, this is not consistent with the reviewer's other statement: "*the excitation properties are related to the far-field*". Also, the reviewer's statement about the "*nonzero mode overlap*" for TE_{31,Γ} and TE_{51,Γ} modes is not true. Since we only study the eigenmodes' properties along the Γ -X high symmetry direction in the k space, they are pure TE mode and hence the mode overlap is zero. This is also mentioned by

the reviewer “Along the high symmetry direction of the k -space, such as Gamma-X for the square lattice the mode polarization in the far-field can be always separated into pure TE and TM.” In addition, the mode overlap is more related to the in-plane waveguide system, and it’s rarely related to the mode excitation in the linear metasurfaces working on the out-of-plane scheme.

To further clarify the mechanism of exciting TE mode with TM polarized source, we have modified the relative text in lines 309~313, pages 15~16 and expanded the discussion in Supplementary Section S6 and reproduced it below for the reviewer’s convenience.

Modified text in lines 309~313, pages 15~16 of the revised manuscript

TE₃₁ and TE₅₁ modes are observed under the excitation of TM polarized light due to the symmetry matching condition: a linear polarized source with one type of symmetry can only excite and couple to the eigenmodes with the same kind of symmetry^[37] (see Supplementary Section S6 for more details).

Supplementary Section S6: Excitation of TE eigenmodes with different polarized sources

The eigenmodes of every two-dimensional PhC can be classified into two distinct polarizations^[8]: transverse electric (TE) modes, in which the electric field is confined to the xy plane (E_x, E_y, H_z), and transverse magnetic (TM) modes, in which the magnetic field is confined to the xy plane (H_x, H_y, E_z). Due to the finite thickness of PhC, the fields become mostly TE-like and TM-like when moving away from the mirror plane of the structure in the z direction. Fig. S6a shows the calculated TE band structure of gap-perturbed ($\Delta L = 30 \mu\text{m}$, $\alpha = 0.5$) PhC. For TE polarized plane wave (Fig. S6c), it carries both electric and magnetic field components in the xy -plane: ($H_x, E_y, 0$) for normal incidence and (H_x, E_y, H_z) for oblique incidence. The situation is similar for TM polarization: ($E_x, H_y, 0$) for normal incidence and (E_x, H_y, E_z) for oblique incidence. The available eigenmodes that can be excited by a certain polarized light depend on the symmetry matching condition between the eigenmodes’ field profiles and the excitation source^[9]. Specifically, as shown in Fig. S6d, under the mirror reflection operation around the x axis, which changes y to $-y$ (noted as σ_y), the magnetic field profiles of TE_{11, Γ} , TE_{21, Γ} , TE_{41, Γ} , and TE_{40, Γ} modes remain the same, showing an even feature. On the contrary, the magnetic field profiles of TE_{31, Γ} and TE_{51, Γ} modes present an odd feature under σ_y operation. Here σ_y is considered because when moving away from Γ to X point, the symmetry group changes from C_{2v} to C_{1h} , and only the representations E and σ_y retain^[10]. Since the magnetic vector of TE (TM) polarized light is even (odd) under σ_y operation, TE₁₁, TE₂₁, TE₄₁, and TE₄₀ (TE₃₁ and TE₅₁) modes are excited and observed in the TE(TM)-polarized transmission spectra (Fig. S6b).

Fig. S6. Excitation of TE eigenmodes with different polarized sources. a, Calculated transverse electric (TE) band structure of gap-perturbed ($\Delta L = 30 \mu\text{m}$, $\alpha = 0.5$) PhCs. The blue, black, and yellow lines represent the bands having GRs, BZF-BICs, and BIC, respectively. The shaded blue area around the GRs shows the linewidth of the modes. b, Calculated angle-resolved transmission spectra of THz-PhC ($\Delta L = 30 \mu\text{m}$) along the Γ -X direction under TE and TM polarizations. c, Schematics of the magnetic and electric vectors configurations of oblique incident TE and TM polarized light. d, Calculated magnetic field profiles H_z of eigenmodes at Γ point for gap-perturbed ($\Delta L = 30 \mu\text{m}$) PhC.

Comment #4

The experimental results are convincing; however, the achieved Q factor is very low because of measurement limitation for THz experiments. Therefore, it is quite unclear what the benefits of "dark BICs" compared to conventional quasi-BICs are in realistic structures - the measured Q factor is strongly reduced by the fabrication losses and setup limitations smearing out any difference.

Response from Authors:

This comment from Reviewer #3 questions the experimental demonstration of the advancements of BZF-BIC over ordinary BICs. It is very similar to Comment #3 from Reviewer #1. Please see the response to Comment #3 from Reviewer #1, in which we have explained in detail and shown revised text and figures. Due to the long length of the response, we have not reproduced it here.

Minor comments from reviewer #3.

Comment #5

The authors noted that there are only three mechanisms leading to BIC formation, which is not true. More mechanisms and their discussion can be found in Ref. [R3].

Response from Authors:

It is true that there are a lot of mechanisms that can be used to engineer BICs. However, most of the mechanisms can be divided into three categories as we described in the manuscript “*mainly three different ways to realize BICs*”. Right in the reference Ref [R3] (pages 2 to 7) mentioned by Reviewer #3, Hsu *et al.* also categorized the BICs into three main types: 1) bound states due to symmetry or separability; 2) bound states through parameter tuning; 3) bound states from inverse construction. In the first type, BICs formed due to symmetry are symmetry-protected BICs, and it’s the first way to realize BICs as described in our manuscript. The separable BICs, where coupling to the continuum is forbidden by separability, are rarely studied and have not been observed experimentally. The second type of BIC formed through parameter tuning is noted as tunable BICs or accidental BICs. According to the parameter tuning condition, accidental BICs can be sub-categorized as Fabry-Pérot BICs, Friedrich-Wintgen BICs, and single-resonance parametric BICs. This is the same as the second way to realize BICs as described in our manuscript. The last type of BIC in Ref [R3] formed through inverse construction is also the same as the third way to realize BICs as described in our manuscript.

To include more details about the three different ways to realize BICs, we have expanded the discussion in lines 48~56, page 3 and reproduced it below for the reviewer’s convenience.

Modified text in lines 48~56, page 3 of the revised manuscript

So far, there are mainly three different ways to realize BICs: i) find eigenmodes that are forbidden to couple to the radiation continuum by symmetry (known as symmetry-protected BICs^[10, 12, 15-16]) or separability (noted as separable BICs^[25-26]); ii) tune a system’s parameters causing destructive interference between several leakage channels (noted as tunable BICs or accidental BICs^[9, 11, 13]). Tunable or accidental BICs can be further sub-characterized as Fabry-Pérot BICs, Friedrich-Wintgen BICs, and single-resonance parametric BICs according to different parameter-tuning scenarios; iii) use inverse construction methods, including potential engineering, hopping rate engineering, and boundary shape engineering^[27].

Comment #6

The authors noted that the conventional BICs are not formed due to band folding which is incorrect. All modes of periodic structures are formed due to bandfolding of waveguide and leaky modes due to a periodic perturbation [see R2].

Response from Authors:

We apologize for the incorrect description. The band folding that we referred is related to the radius and gap perturbations studied in this work. The conventional BIC $TE_{40,\Gamma}$ exists even without radius and gap perturbation so we stated that “bright BICs that are formed without band

folding". To avoid any further confusion, we have modified the text in lines 84~88, page 4, and reproduced it below for the reviewer's convenience.

Modified text in lines 84~88, page 4 of the revised manuscript

As distinct from conventional BICs that do not experience band folding caused by a gap or radius perturbations and show rapid Q -factor-decay, BZF-BICs show perturbation-dependent enhancement of ultrahigh Q factor in the large portion of the momentum space, as conceptually illustrated in Fig. 1a.

Comment #7

The authors could add experimentally measured dependence of Q factor on the angle of incidence or k_x to compare it with the calculated dependencies in Figure 2.

Response from Authors:

We thank the reviewer for the suggestion. Although the evolution feature of GRs, BZF-BICs, and ordinary BICs' radiation loss in the momentum space can be observed from the measured angle-resolved transmission spectra (Fig. 5 in the main text), k_x -resolved Q factor would definitely provide a clearer picture and help readers understand the Q factor evolution in a quantitative way. However, we are currently limited by the measurement setup and the Q factors measured have discrepancies with simulated ones at oblique incidence. As shown in Fig. S9a, the observed quasi-BZF-BIC TE₅₁ mode of the PhC ($\Delta L = 22 \mu\text{m}$) at normal incidence (black line) indicates that the incident terahertz beam has a convergence angle ($\theta \approx 6^\circ$), so the source has a spread of k points which excite the quasi-BZF-BIC mode even at normal incidence and thus lower the measured Q factor. The measured Q factors are nearly an order of magnitude smaller than the simulated ones at a small incidence angle (Fig. S9c). In addition, the measured resonance's amplitude decreases as the incident angle increases from 0 due to the degraded collection efficiency of the measurement setup at oblique incidences. As shown in Fig. S9d, the measured and simulated Q factors of GR TE₃₁ mode are close at normal incidence. As the incident angle increases, the measured TE₃₁ mode's amplitude decreases, and discrepancy appears between measured and simulated Q factors.

Since the measured Q factors at oblique incidence have a large discrepancy with simulation results and do not show the real evolution feature, the measured Q factors at different k_x values and different gap perturbations are not shown in the main text to avoid any confusion. However, we have added discussions about the limitation of our setup on the oblique incident measurements in the main text and Supplementary Section S9. The relative contents have been reproduced below for the reviewer's convenience. We should note that the limitations of our measurement setup will not affect the main idea of this work that quasi-BZF-BICs have robustness and reduced radiation loss (enhanced Q factor) in the momentum space, which is shown in the measured angle-resolved transmission spectra.

Modified text in lines 389~392, page 19 of the revised manuscript

In addition, the amplitude decreases at oblique incidence due to the degraded collection efficiency of the measurement setup (see Supplementary Section S9), which lowers the measured Q factor at large incident angles.

Supplementary Section S9: Limitations of measurement setup

A maximum Q factor of 860 was measured in this work. However, higher Q factor measurement is difficult due to the following limitations: 1) the resolution of our terahertz spectroscopy is 0.58 GHz, which implies that the maximum Q factor that we could measure is ~ 1000 if we consider a resonant frequency of 0.6 THz; 2) the diameter of the terahertz beam spot is 8 mm, so the excited mode has a finite lateral size of $L \approx 8$ mm. This finite-sized mode consists of a spread of k points with $\delta k_{\text{mode}} \approx 2\pi/L \approx 3.5 \times 10^{-2}$ ($2\pi/a_1$); 3) the terahertz beam has a convergence angle $\theta \approx 6^\circ$, so the source also has a spread of k points with $\delta k_{\text{source}} \approx (2\pi/\lambda) \sin(\theta) \approx 5.2 \times 10^{-2}$ ($2\pi/a_1$). As shown in Fig. S9a, quasi-BZF-BIC TE_{51} mode of THz-PhC ($\Delta L = 22$ μm) is observed at normal incidence (black line), indicating that the normal incident terahertz beam carries non-zero k components. The measured radiative loss will be the averaged value within this spread of k points. In addition, as the incident angle increases, the measured TE_{31} mode's amplitude decreases due to the degraded collection efficiency of the measurement setup at oblique incidences. It leads to a large discrepancy between the measured and simulated Q factors at large incident angles (Fig. S9c and S9d).

Fig. S9. The discrepancy between measured and simulated results. a, Measured and b, simulated transmission spectra of THz-PhC ($\Delta L = 22$ μm) at different incident angles. Measured and simulated Q factors of c, TE_{51} and d, TE_{31} mode.

References.

[R1] Murai, S., Abujetas, D. R., Liu, L., Castellanos, G. W., Giannini, V., Sánchez-Gil, J. A.,

Tanaka, K., Gómez Rivas, J., Engineering Bound States in the Continuum at Telecom Wavelengths with Non-Bravais Lattices. Laser Photonics Rev 2022, 16, 2100661.

ArXiv arXiv:2204.09986v1

[R2] Fan, S. and Joannopoulos, J.D., 2002. Analysis of guided resonances in photonic crystal slabs. Physical Review B, 65(23), p.235112.

[R3] Hsu, C.W., Zhen, B., Stone, A.D., Joannopoulos, J.D. and Soljačić, M., 2016. Bound states in the continuum. Nature Reviews Materials, 1(9), pp.1-13.

References

[C1] Overvig, Adam C., et al. "Selection rules for quasibound states in the continuum." *Physical Review B* 102.3 (2020): 035434.

[C2] Overvig, Adam C., Sajan Shrestha, and Nanfang Yu. "Dimerized high contrast gratings." *Nanophotonics* 7.6 (2018): 1157-1168.

[C3] Chen, Zihao, et al. "Observation of miniaturized bound states in the continuum with ultra-high quality factors." *Science Bulletin* 67.4 (2022): 359-366.

[C4] Lee, Jeongwon, et al. "Observation and differentiation of unique high-Q optical resonances near zero wave vector in macroscopic photonic crystal slabs." *Physical Review Letters* 109.6 (2012): 067401.

[C5] Joannopoulos, John D., et al. "Photonic crystals: molding the flow of light" Princet. Univ. Press. Princeton, NJ (2008).

Reviewers' Comments:

Reviewer #1 (Remarks to the Author):

In this work, the authors propose a new type of BIC named dark BIC, which originated from the Brillouin zone folding by periodic perturbation of guided modes. Without perturbation which introduces the band folding, these guided modes are outside the light cone and exhibit an infinite Q factor. When the perturbation is small, the guide modes have a high Q factor. Dark BICs inherit both the high-Q factor from the guided modes and the symmetry-protected BICs, and thus exhibit ultrahigh-Q factors. Compared to the rapid Q-factor-decay feature of isolated BICs (so-called bright BIC) in previous work, the dark BIC presents a perturbation-dependent dramatic enhancement of Q factors even far from the topological center. Besides, the authors provide an experimental demonstration of dark BICs in the sub-THz region. Generally speaking, the manuscript is well-written, and the content is trustworthy. The idea of introducing band-folded guided modes to further increase the Q factor of the dark BICs is quite interesting. However, I cannot recommend the paper in its current form. I will explain my concerns below.

Like merging BICs, dark BICs should be robust against perturbations and randomness. Perturbation over α is partly explored, while randomness is not investigated.

On line 149, the authors claim that $\Delta L=1\mu\text{m}$ is a large perturbation, which might be true for the sub-THz experiments. However, for more popular optical systems, $\Delta L=1\mu\text{m}$ is only around 0.3% error and far beyond the capabilities of most facilities. This is related to the claims in the abstract, lines 19, 25, the values "nine billion", "six orders" are vague. The values should be presented more carefully.

For the THz experiments presented in this work, the Q factor is quite small (~ 400) and the capabilities of dark BICs are not fully resolved. To be more specific, the experiments fail to show the advantages of dark BICs (far from the claims in the abstract. At least should be orders higher than other bright BICs as claimed by the authors). Meanwhile, the $1/\alpha^2$ behavior in Fig. 4e is not good (numerical results are perfect).

The critical feature of dark BIC is the slow decay of the Q factor far from the topological center. In this manuscript, the quadratic decay feature $Q=Q_0/(\alpha^2 k^2)$ was provided but lack of proof. I understand it is proven in some other references, but a simple and intuitive demonstration might be better for the readers to follow. Meanwhile, to what extent does the inverse-square law work well? Say, still valid if α approaches 1?

The statement in line 74, "which eventually diverges to infinity in the whole momentum space as the perturbation approaches zero" seems kind of misleading. The eigenstates in the case of zero perturbation are guided modes rather than BICs due to the lack of continuum.

Reviewer #2 (Remarks to the Author):

In this work, the authors experimentally demonstrate BICs and quasi-BICs using a supercell method to increase the robustness of Q-factor with regards to in-plane wavevector ($k_{||}$) compared to using symmetry protected BICs in the first Brillouin zone, these folded modes can achieve improved Q-factors. I think the robustness is neat, but these modes have previously been proposed in theory (doi: 10.1103/PhysRevB.102.035434) and experimentally demonstrated (doi: 10.1038/s41377-022-00905-6, and 10.1515/nanoph-2020-0375). But the characterization of robustness is new and besides the the previously mentioned issues the manuscript is well thought out and organized. Because of this, I think the manuscript requires significant revisions to convey the advancements made more specifically and precisely. Along these lines I have the following comments:

1. The notation of bright and dark BICs is not consistent with the literature and can create confusion. I think the notation used by Overvig et al. (reference [28] in the manuscript) and table one under the mode notation are more straightforward.
2. The process for folding BICs from outside the first Brillouin zone (FBZ) to the FBZ needs to more clearly and explicitly cite previous work done by Overvig et al. (reference [28] in the manuscript) who generated selection rules for all the methods for folding BICs into the FBZ.
3. For the THz design these structures are large, and I would have imagined high Q-factors could be achieved, but only a Q-factor of 400 is observed. What is the cause? In the NIR previous work has demonstrated greater than 600 using the BIC Brillouin zone folding technique (doi: 10.1515/nanoph-2020-0375).

Reviewer #3 (Remarks to the Author):

In the manuscript, "Dark bound states in the continuum metasurfaces", Wenhao Wang et al. investigate numerically and experimentally resonant optical modes in dielectric metasurfaces composed of equally spaced meta-atoms. The authors study the evolution of resonant properties of modes after band folding associated with dimerizing of meta-atoms and period doubling. They demonstrate that the guided modes (GM) transform into quasi-guided modes which possess a very large radiative Q factor and can become bound states in the continuum (BICs) at the Gamma point depending on the type of perturbation leading to dimerization of the unit cell. The authors investigate numerically how the perturbation type affects the formation of BICs and quasi-GMs, their topological charge and far-field polarization structure. The quasi-GMs with and without BIC features are demonstrated experimentally for the perturbation changing the distance between adjacent meta-atoms. The authors show numerically and experimentally that the Q factor of GMs both with and without BICs at the Gamma point is very robust to the angle of incidence compared to conventional quasi-BICs. The highest measured Q factor of BICs is below 400. The authors associate the predicted GMs with BIC feature as a new type of resonance called "dark BIC".

The idea of engineering of ultrawideband high-Q resonances in dielectric metasurfaces via band folding and dimerizing of meta-atoms is timely and is of high fundamental and practical significance for the field of photonics and optics of subwavelength metastructures. The numerical results of the manuscript are mostly of high technical quality and are supported by solid experimental studies. The paper would make a great candidate for publication in Nature Communications, however, it strongly lacks novelty - an identical study was published earlier this year [see Ref. R1] demonstrating the same effect in dielectric Si metasurfaces in the optical and near-IR range, supported by profound theoretical analysis and measurements. To my regret, all the core ideas of this manuscript, including the physics of "dark BICs", were explored in Ref. [R1], moreover, the Q factor achieved in [R1] is more than 1000 in the near-IR range which exceeds the value achieved in the current manuscript in the THz range. Therefore, I may conclude that now when the physics and many of essential details of the paper are well understood from the previous comprehensive studies, the manuscript lacks substantial scientific novelty that would make it acceptable for publication in Nature Communications. Also, there are some substantial technical flaws in the presented results, that do not affect the main idea but could confuse the readers. Below, I outline a detailed list of comments of criticisms.

Major comments.

The core idea of the manuscript is the engineering of high-Q resonances within the whole k-space by band folding of the metasurface resonant modes via dimerization of meta-atoms. A very recent publication [see Ref. R1], which appeared online on 26 Aug 2022 (preprint available on arXiv from 21 Apr 2022), explored the very same idea in an identical structure of Si disks arranged in a

square lattice metasurface. The authors of Ref. [R1] did not use the term "dark BIC", but they explored how the band folding effect transforms GM modes into quasi-GMs with a giant Q factor. Moreover, the paper [R1] explores how BICs are formed at the Gamma point for the quasi-GMs achieved through band folding depending on the type of dimerizing perturbation - the authors in [R1] study the same "gap" and "radius" perturbations as in the current manuscript. The reason of why BICs are formed for one quasi-GMs for "gap" perturbation, and for other quasi-GMs for "radius" perturbations are studied in detail in Ref. [R1] with a comprehensive theory behind that. Moreover, Ref. [R1] is an experimental study which confirms all discussed effects in the near-IR and visible range and shows that the Q factor can exceed 1000 at ~ 1240 nm and 300 at ~ 580 nm.

The terminology "dark BIC" used by the authors is confusing both physically and mathematically. All modes appearing due to band folding of GMs represent quasi-GMs (or, guided resonances [R2]) due to finite radiation losses. Then, at the Gamma point some of them are BICs with infinite Q factor. Away from the Gamma point such modes represent conventional quasi-BICs. The reason that these quasi-BICs possess a large Q factor in the whole Brillouin zone is due to band folding but not due to the BIC physics - the other GMs without BICs also possess a very high Q factor in the whole k-space. Thus, the observed states are not BICs anywhere except the Gamma point and the core physics behind the high Q factor is not due to BIC nature. Moreover, the BICs are already dark modes, so calling them dark is confusing.

I am very confused by the statement that TM polarized excitation can excite TE modes in metasurface made by the authors and explained in the Supporting Information. Along the high symmetry direction of the k-space, such as Gamma-X for the square lattice the mode polarization in the far-field can be always separated into pure TE and TM. This separation can achieve both for the eigenmodes and the excitation beam. This can be proven if the quasi-guided modes are decomposed into the basis of guided modes of an effective waveguide: the coupling between a TE and TM waveguide modes is proportional to their mode overlap within the unit cell. It is nonzero only for directions in the k-space that are not aligned with the lattice axes. Therefore, the linearly TE (TM) polarized excitation source cannot excite TM (TE) modes. The explanation made in Supporting Information is very confusing, e.g. it states that TE eigenmodes have a nonzero H_z component of the field. It is true for the near-field, but the excitation properties are related to the far-field and the out-of-plane components of all fields are zero for plane waves. The authors should specify why they observe TE modes for TM excitation in calculations and experiment. Also, it is confusing that TE₃₁ and TE₅₁ modes are not seen for TE excitation - there should be a nonzero signature of them in any case because of nonzero mode overlap.

The experimental results are convincing; however, the achieved Q factor is very low because of measurement limitation for THz experiments. Therefore, it is quite unclear what the benefits of "dark BICs" compared to conventional quasi-BICs are in realistic structures - the measured Q factor is strongly reduced by the fabrication losses and setup limitations smearing out any difference.

Minor comments.

The authors noted that there are only three mechanisms leading to BIC formation, which is not true. More mechanisms and their discussion can be found in Ref. [R3].

The authors noted that the conventional BICs are not formed due to band folding which is incorrect. All modes of periodic structures are formed due to bandfolding of waveguide and leaky modes due to a periodic perturbation [see R2].

The authors could add experimentally measured dependence of Q factor on the angle of incidence or k_x to compare it with the calculated dependencies in Figure 2.

References.

[R1] Murai, S., Abujetas, D. R., Liu, L., Castellanos, G. W., Giannini, V., Sánchez-Gil, J. A., Tanaka, K., Gómez Rivas, J., Engineering Bound States in the Continuum at Telecom Wavelengths with Non-Bravais Lattices. *Laser Photonics Rev* 2022, 16, 2100661.

ArXiv arXiv:2204.09986v1

[R2] Fan, S. and Joannopoulos, J.D., 2002. Analysis of guided resonances in photonic crystal slabs. *Physical Review B*, 65(23), p.235112.

[R3] Hsu, C.W., Zhen, B., Stone, A.D., Joannopoulos, J.D. and Soljačić, M., 2016. Bound states in the continuum. *Nature Reviews Materials*, 1(9), pp.1-13.

Brillouin Zone Folding Driven Bound States in the Continuum